# SARS-CoV-2 Rapidly Infects Peripheral Sensory and Autonomic Neurons, Contributing to Central Nervous System Neuroinvasion before Viremia

**DOI:** 10.3390/ijms25158245

**Published:** 2024-07-28

**Authors:** Jonathan D. Joyce, Greyson A. Moore, Poorna Goswami, Telvin L. Harrell, Tina M. Taylor, Seth A. Hawks, Jillian C. Green, Mo Jia, Matthew D. Irwin, Emma Leslie, Nisha K. Duggal, Christopher K. Thompson, Andrea S. Bertke

**Affiliations:** 1Translational Biology, Medicine, and Health, Virginia Polytechnic Institute & State University, Blacksburg, VA 24060, USA; jjoyce84@vt.edu (J.D.J.);; 2Center for Emerging Zoonotic and Arthropod-borne Pathogens, Virginia Polytechnic Institute & State University, Blacksburg, VA 24060, USA; 3Biomedical and Veterinary Science, Virginia Maryland College of Veterinary Medicine, Virginia Polytechnic Institute & State University, Blacksburg, VA 24060, USA; 4Population Health Sciences, Virginia Maryland College of Veterinary Medicine, Virginia Polytechnic Institute & State University, Blacksburg, VA 24060, USA; 5Biomedical Sciences and Pathobiology, Virginia Polytechnic Institute & State University, Blacksburg, VA 24060, USA; 6School of Neuroscience, Virginia Polytechnic Institute & State University, Blacksburg, VA 24060, USA

**Keywords:** SARS-CoV-2, COVID-19, neuroinvasion, trigeminal ganglia, superior cervical ganglia, dorsal root ganglia, peripheral nervous system, autonomic nervous system, allodynia, neuropilin-1

## Abstract

Neurological symptoms associated with COVID-19, acute and long term, suggest SARS-CoV-2 affects both the peripheral and central nervous systems (PNS/CNS). Although studies have shown olfactory and hematogenous invasion into the CNS, coinciding with neuroinflammation, little attention has been paid to susceptibility of the PNS to infection or to its contribution to CNS invasion. Here we show that sensory and autonomic neurons in the PNS are susceptible to productive infection with SARS-CoV-2 and outline physiological and molecular mechanisms mediating neuroinvasion. Our infection of K18-hACE2 mice, wild-type mice, and golden Syrian hamsters, as well as primary peripheral sensory and autonomic neuronal cultures, show viral RNA, proteins, and infectious virus in PNS neurons, satellite glial cells, and functionally connected CNS tissues. Additionally, we demonstrate, in vitro, that neuropilin-1 facilitates SARS-CoV-2 neuronal entry. SARS-CoV-2 rapidly invades the PNS prior to viremia, establishes a productive infection in peripheral neurons, and results in sensory symptoms often reported by COVID-19 patients.

## 1. Introduction

Up to 80% of people infected with SARS-CoV-2, the virus responsible for COVID-19, report neurological symptoms. Although some of these symptoms implicate central nervous system (CNS) involvement, many COVID-related symptoms indicate that SARS-CoV-2 impacts the peripheral nervous system (PNS), including sensory and autonomic systems [1,2,3]. Both central and peripheral symptoms, such as fatigue, memory issues, “brain fog”, hyper/hypoesthesia, and autonomic dysfunction, can persist as part of post-COVID-19 syndrome (“long COVID”) long after acute infection [4]. Detection of the virus, viral RNA, and antigens in the cerebrospinal fluid and brains of COVID-19 patients indicates SARS-CoV-2 is neuroinvasive, which has also been documented for common cold and epidemic coronaviruses (HCoV-OC43, HCoV-229E, MERS, SARS) [5,6,7,8,9,10,11,12,13]. Given that neuroinvasive viruses can reach the PNS/CNS via invasion of nerve terminals or via dissemination through the blood, studies early in the pandemic attempted to determine which of these mechanisms is used by SARS-CoV-2.

As anosmia is a primary symptom of COVID-19, early studies assessed CNS invasion via olfactory sensory neurons (OSNs) using transgenic mice expressing human angiotensin converting enzyme 2 (hACE2), the receptor for SARS-CoV-2 [14,15,16]. Virus detection in OSNs and sustentacular cells confirmed these cells as viable routes into the CNS. However, the oronasal mucosa and target organs of SARS-CoV-2 (lungs, gut) are heavily innervated by sensory and autonomic neurons of the PNS, providing an alternative route of entry directly into peripheral neurons from the mucosa rapidly upon infection, which could facilitate invasion into functionally connected CNS tissues. However, little attention has been paid to the role of the PNS during infection and to its potential as an alternative route of CNS invasion. Since PNS symptoms are often reported among non-hospitalized COVID-19 patients, who constitute the majority of those infected, and up to one-third of those with post-COVID-19 syndrome, understanding the susceptibility of the PNS to SARS-CoV-2 infection and the functional consequences thereof is essential [17,18].

Therefore, we assessed the susceptibility of PNS sensory (trigeminal ganglia-TG, lumbosacral dorsal root ganglia-LS-DRG) and autonomic (superior cervical ganglia-SCG) neurons to infection with SARS-CoV-2 following intranasal inoculation of K18-hACE2 transgenic mice (hACE2 mice), wild-type C57BL/6J mice (WT), and golden Syrian hamsters. To determine if infection of the PNS preceded and contributed to CNS infection, we assessed early neuroinvasion of CNS tissues functionally connected to peripheral neurons innervating oronasal and gut mucosa (lumbosacral spinal cord and specific brain regions [olfactory bulb, cortex, hippocampus, brainstem, cerebellum]). We also characterized viral growth kinetics in primary PNS neuronal cultures and investigated the contribution of neuropilin-1 (NRP-1) to entry into PNS neurons.

We focused on early neuroinvasion of ancestral SARS-CoV-2, as differences in neurological symptoms between variants are only beginning to be reported. A recent epidemiological study has shown the odds ratio for developing post-COVID-19 syndrome following infection with ancestral SARS-CoV-2 is 4.0, whereas the odds ratio following Omicron infection is 0.18 [19]. While investigation of the neuroinvasive potential of contemporary SARS-CoV-2 variants (e.g., Omicron) is worthwhile and will be the focus of our future studies, the majority of the estimated 79.2 million COVID-19 cases identified in 2020 were driven by infection with the Wuhan or WA1/2020 strain [20,21,22]. These cases occurred at a time before antivirals, monoclonal antibody therapies, and vaccines were widely available to prevent or ameliorate infection and presumably limit PNS/CNS disease [23]. It is this cohort of COVID-19 cases, having occurred before the alpha, beta, gamma, delta, and omicron waves of 2021, that provides the greatest longitudinal dataset for our current understanding of the long-term impacts of neuroinvasive SARS-CoV-2 infection [21].

We show that SARS-CoV-2 productively infects PNS sensory neurons and that autonomic neurons, while susceptible to infection, sustain significant cytopathology and do not release infectious virus. Recovery of infectious virus from the TG, which innervates oronasal mucosa and extends central projections into the brainstem, suggests an alternative route of CNS neuroinvasion independent of OSNs. We also recovered infectious virus from LS-DRGs and the LS spinal cord, which may underlie some sensory disturbances experienced in COVID-19. We further show that SARS-CoV-2 is capable of replicating in specific brain regions, with the highest virus concentration detected in regions functionally connected to the PNS. Furthermore, invasion of the PNS and functionally connected regions of the CNS occurs rapidly after infection, before viremia, strongly suggesting that direct entry and transport into the CNS occurs through PNS infection. We also show that neuronal invasion can be partially inhibited by pharmacological blockade of neuropilin-1 in vitro. In summary, neuroinvasion of SARS-CoV-2 occurs rapidly via direct peripheral neuronal entry, can be independent of hACE2, and is facilitated by NRP-1.

## 2. Results

### 2.1. Clinical Data

To assess the susceptibility of PNS and CNS neurons to SARS-CoV-2 infection, we intranasally (IN) inoculated hACE2 and WT mice with 10^3^ or 10^5^ PFU SARS-CoV-2 USA-WA1/2020 (*n* = 12/group; Appendix A). Animals were monitored daily and tissues collected three and six days post-infection (dpi). Weight loss began 3 dpi and death occurred 6 dpi (14%) in hACE2 mice inoculated with 10^5^ PFU (Appendix A). While only one hACE2 mouse in the 10^5^ PFU inoculated group succumbed to infection during our study, infection likely would have been universally fatal at later timepoints, as significant mortality is commonly reported in hACE2 mice beginning 5–6 dpi. RT-qPCR showed low-level transient viremia and lung infection at 3 and 6 dpi (Appendix A). These results are similar to previous studies and demonstrate successful infection [16,24,25].

### 2.2. PNS Sensory Trigeminal Ganglia (TG) and Sympathetic Superior Cervical Ganglia (SCG) Neurons Innervating the Oronasopharynx Are Susceptible to Infection

Both sensory and sympathetic pathways through TGs and SCGs, respectively, could serve as neural routes for neuroinvasion of the CNS. Trigeminal nerves provide sensory innervation to oronasal mucosa, projecting their central processes into the brainstem. SCGs provide sympathetic innervation to salivary and lacrimal glands and vasculature of the head and brain, with preganglionic neurons residing in the spinal cord. Thus, sensory and sympathetic nerve endings in the oronasal mucosa provide a direct pathway into the CNS. To assess susceptibility of these peripheral neurons to infection with SARS-CoV-2, TGs and SCGs were assessed for viral RNA, proteins, and infectious virus. We detected viral RNA at 3 and 6 dpi in TGs and SCGs in both inoculum groups in hACE2 and WT mice. Viral RNA concentrations were lower in WT than hACE2 mice (Figure 1a,b) but increased over time in TGs and SCGs of both, suggesting genome replication. Immunostaining detected nucleocapsid (SARS-N) in ≈41% of TG neurons of hACE2 mice and ≈37% of TG neurons of WT mice (Figure 1c,d and Appendix A). Nearly all SCG neurons from hACE2 mice (≈97%) were SARS-N-positive (Figure 1c), showing substantial pathology with vacuolated neurons and loss of ganglionic architecture (Figure 1d). Although SCGs from WT mice remained intact, SARS-N was evident in ≈93% of neurons, and some vacuolization was observed. Immunostaining detected spike (SARS-S) in both ganglia similar to detection of SARS-N (Figure 1e and Appendix A). Viral genome replication was confirmed by double-stranded RNA (dsRNA) immunostaining in both ganglia (Figure 1f and Appendix A). Infectious virus was recovered in TGs from one 10^5^ PFU-inoculated hACE2 mouse at 3 dpi (5 PFU/mg homogenate) and another at 6 dpi (2 log PFU/mg homogenate). Infectious virus was not detected in SCGs. The pathology of the ganglia suggests SARS-CoV-2 may have produced such significant cytotoxicity that production of viral progeny was impossible. The use of multiple complementary assays indicate that viral RNA and protein can be isolated from both TGs and SCGs of hACE2 and WT mice and that infectious virus can be recovered from TGs. Thus, TGs and SCGs are susceptible to infection with SARS-CoV-2, TGs may serve as a route of CNS invasion, and neuroinvasion can occur independently of hACE2.

### 2.3. PNS Sensory Lumbosacral Dorsal Root Ganglia (LS-DRG) and CNS Lumbosacral Spinal Cord (LS-SC) Neurons Are Susceptible to Infection

Extending our investigation beyond PNS innervations of the oronasopharynx, we assessed the LS-DRG and LS-SC as above. The LS-DRG conveys sensory information (pain, pressure, position) from the periphery and internal organs to the LS-SC. Similar to our results from TGs, we detected viral RNA at 3 and 6 dpi in LS-DRGs in both inoculum groups in hACE2 and WT mice (Figure 2a). RNA concentrations increased over time, suggesting genome replication. Similar results were found in the LS-SC (Figure 2b). Immunostaining demonstrated SARS-N in ≈42% of LS-DRG neurons of hACE2 mice and ≈24% of WT neurons (Figure 2c,d and Appendix A). Immunostaining for SARS-S in LS-DRGs was similar to that of SARS-N, with SARS-S present throughout hACE2 LS-DRGs, and also in satellite glial cells (SGCs) (Figure 2e,g and Appendix A). Immunostaining for dsRNA mirrored that of SARS-N and SARS-S, indicating viral genome replication (Figure 2f and Appendix A). Infectious SARS-CoV-2 (2 PFU/mg homogenate) was recovered from the LS-DRG homogenate of one hACE2 mouse at 6 dpi, verifying productive infection. Presence of viral RNA and infectious virus in neurons with no axonal targets in the head or lungs suggested spread via hematogenous dissemination or via axonal transport from a distal site of infection. Given that LS-DRGs project to the LS-SC, and transport from the LS-DRG to the LS-SC (or vice versa) is possible, the LS-SC was also immunostained. We identified punctate SARS-N staining inside LS-SC neurons, as well as a diffuse SARS-N signal throughout the LS-SC (Figure 2h and Appendix A) suggestive of free virus, or at least N protein, in the LS-SC. Immunostaining for microglia (Iba1, Figure 2i) and astrocytes (S100b, Figure 2j) showed minimal co-localization between SARS-N and these cell markers. Infectious virus (4 log PFU/mg homogenate) was recovered from the LS-SC of one hACE2 mouse at 6 dpi, demonstrating productive infection. These data further demonstrate that PNS sensory neurons, as well as functionally connected neurons in the spinal cord, are susceptible to infection with SARS-CoV-2.

### 2.4. Individual Brain Regions Support Varying Levels of Viral Invasion and Reproduction

To determine if SARS-CoV-2 was present in specific brain regions with functional connections to PNS ganglia, we assessed the olfactory bulb, hippocampus, cortex, brainstem, and cerebellum. Previous studies have tested brain homogenates to assess viral RNA and infectious virus in the brain, which does not allow for spatial analysis and may obscure detection of low levels of virus in specific brain regions [14,16,26,27,28]. We detected viral RNA in all brain regions at 3 dpi, which increased by 6 dpi, in both hACE2 and WT mice (Figure 3a–e). Higher viral RNA concentrations were found in hACE2 mice compared to WT mice in all regions, with the highest in the brainstem and hippocampus at 6 dpi. In WT mice, all regions contained similar quantities of SARS-CoV-2 RNA, suggesting that ACE2-independent spread or replication through the nervous system may differ from hACE2 mice. Infectious virus was recovered from the hippocampus and brainstem (3 log PFU/mg homogenate) as early as 3 dpi in hACE2 mice, supporting viral spread into the CNS from the PNS (Figure 3f). Unexpectedly, infectious virus was also recovered from the hippocampus (10^3^ group: 2 PFU/mg homogenate; 10^5^ group: 5 PFU/mg homogenate) and brainstem (10^5^ group: 3 PFU/mg homogenate) from WT mice at 6 dpi. These results indicate that viral invasion and replication in the brain is not uniform, that the highest concentrations of virus are found in areas connected to the trigeminal and limbic systems, and that neuroinvasion is not solely dependent on hACE2 since the WT mice do not express this receptor. To further assess localization of SARS-CoV-2 in brain regions, we immunostained sagittal sections of hACE2 and WT brains for SARS-N. At 6 dpi, immunostaining of hACE2 brains revealed extensive neuroinvasion into brain regions linked to our target PNS ganglia, notably the brainstem, which receives input from the TG, suggesting axonal transport (Appendix A). Immunostaining of WT brains was unremarkable (Appendix A), as previously published. Nevertheless, viral RNA and the infectious virus were readily recoverable throughout WT brains, underscoring the value of using multiple complementary assays to confirm the presence of a neuroinvasive virus.

### 2.5. SARS-CoV-2 Productively Infects Primary Cultured PNS Sensory but Not Autonomic Neurons from Adult Mice

To confirm that PNS sensory and autonomic neurons are both susceptible and permissive to infection with SARS-CoV-2, resulting in release of infectious virus, and to establish basic replication kinetics of SARS-CoV-2 in PNS neurons, primary neuronal cultures were established from LS-DRGs, TGs, and SCGs from 8–10-week-old hACE2 and WT mice. Following infection, media and cells were analyzed separately for viral RNA and infectious virus to differentiate between intracellular replication and release of infectious virus. Viral RNA levels increased in SCG neurons of hACE2 and WT mice, although the increase was mostly in cells, not media (Figure 4a). Infectious virus was not detected in cellular or media fractions from SCG cultures (Figure 4b). While viral genome replication occurred in SCGs, infectious virus was not released, suggesting abortive infection in sympathetic autonomic neurons. Considering the significant pathology of the SCGs in vivo, SARS-CoV-2 appears to be cytotoxic to SCG neurons prior to production of viral progeny.

In sensory neurons, viral RNA increased in TG and LS-DRG neurons in cyclical 48 h intervals in both cellular and media fractions, with the exception of WT LS-DRG neurons, in which RNA steadily increased, ultimately reaching concentrations similar to hACE2 neurons (Figure 4a). Similarly, infectious virus was recovered cyclically at 48 h intervals from TG and LS-DRG neurons and media from hACE2 mice, but only a single cycle at 2 dpi in WT neurons (Figure 4b). These data indicate that successive rounds of genome amplification and infectious virus release occur in sensory neurons of the TG and LS-DRG, either within individual neurons without killing them or in additional neurons through a second infection cycle, although sustained production of viral progeny is dependent on hACE2. Dependence on hACE2 is consistent with neuron-to-neuron spread and replication in additional neurons producing the successive infectious virus release.

In parallel, primary neuronal cultures were infected and fixed at 1, 2, and 3 dpi for SARS-N immunostaining. In addition to genome replication, SCG neurons were permissive for SARS-N protein expression, shown by positive immunofluorescence in both hACE2 (Figure 4c and Appendix A) and WT neurons (Figure 4d and Appendix A), with activated SGCs accompanying dying neurons by 3 dpi (Figure 4c,d). In LS-DRG neurons, several phenotypes were observed in both hACE2 (Figure 4c and Appendix A) and WT neurons (Figure 4d and Appendix A), including perinuclear SARS-N staining (1 dpi) and punctate staining in the cytoplasm of enlarged neurons (3 dpi), likely representing replication compartments (Appendix A). Infected LS-DRGs (Figure 4e) showed a variety of phenotypes, including loss of membrane integrity (inset 1), cytoplasmic puncta (inset 2), and seemingly healthy neurons strongly expressing SARS-N in cytoplasm and processes (inset 3). Infected SGCs, which appeared to be activated, were also present in primary neuronal cultures (Figure 4e inset 3, yellow arrowheads). The modest increase in viral RNA and infectious virus in media (Figure 4a,b) can be explained by the small percentage of LS-DRG neurons (~5% hACE2, ~2% WT) and SCG neurons (<1% hACE2 or WT) that became productively infected in culture (Figure 4f). Taken together, these data show that PNS sensory neurons are both susceptible and permissive to SARS-CoV-2 infection, resulting in release of infectious virus. While autonomic neurons are susceptible to infection with SARS-CoV-2, and genome replication can occur, they are not permissive to release infectious virus.

### 2.6. Neuroinvasion of the PNS and CNS Occurs before Viremia

To determine if neuroinvasion is driven by hematogenous entry or direct neuronal entry, PNS and CNS tissues were assessed 18 and 42 hpi after intranasal inoculation of hACE2 and WT mice (Figure 5a,b). By 18 hpi, although no viral RNA was detected in blood at this time point, viral RNA was detected in all PNS ganglia and the majority of functionally connected CNS tissues in both hACE2 and WT mice. Additionally, viral RNA was present in the salivary glands, innervated by SCG.

By 42 hpi, viremia was only detected in a single hACE2 mouse, and viral RNA had increased in all hACE2 PNS and CNS tissues except SCGs. In WT mice, viremia was not detected, and viral RNA was no longer detected in salivary glands, SCGs, or LS-DRGs but had increased in some brain regions by 42 hpi. Immunostaining did not detect SARS-N in any tissues, indicating the virus was transiting through PNS tissues when collected but had not yet begun replication or was below the level of detection for immunostaining.

To verify that these results were reproducible in an alternative non-transgenic animal model, golden Syrian hamsters were infected (*n* = 9) and tissues collected 18 hpi, 42 hpi, and 3 dpi (Figure 6a–c, Appendix A). Consistent with mice, viral RNA was detected in all PNS and CNS tissues at 18 hpi, preceding viremia, with a continual increase at 42 hpi, indicating direct neural invasion independent of viremia (Figure 6a,b). Unlike in mice, immunostaining showed the presence of SARS-N in all PNS ganglia at 18 hpi, with continued presence through 3 dpi. Substantial vacuolization was observed in the SCGs at 3 dpi, similar to the pathology observed in mice (Figure 6c). SARS-N staining was detected in several neurons in infected hamster brains (Appendix A).

In light of the LS-DRG/spinal cord infection observed in mice, the functional impact of infection on the sensory nervous system was assessed. Using the von Frey assay, a significant decrease (*p* = 0.0001) in the amount of pressure required to elicit a withdrawal reflex was noted, demonstrating allodynia (Figure 6d). Allodynia occurred rapidly as a consequence of infection; 55% (*n* = 5 of 9) of hamsters demonstrated allodynia by 18 hpi, and all remaining hamsters (*n* = 3 of 3) by 3 dpi. Thus, neuroinvasion occurs rapidly after infection, is mediated by invasion of and transport along neurons, can occur independent of hACE2, and functionally impacts PNS sensory neurons, resulting in allodynia. Other studies have shown transcriptional changes in the PNS (TG), as early as 3 dpi following SARS-CoV-2 infection of hamsters [29]. Our results show neuroinvasion via direct neuronal entry into the SCG, TG, and LS-DRG within 18 h following intranasal infection, providing a physiological mechanism for neuroinvasion and potentially for the previously reported dysregulation of sensory neuronal gene expression.

### 2.7. Neuronal Entry in the PNS Involves Neuropilin-1 (NRP-1)

NRP-1 has been shown to interact with SARS-S, thereby enhancing viral binding/entry in non-neuronal cells [30,31,32]. Since our WT mice and hamsters were infected despite the absence of hACE2, and since results from studies using non-neuronal cells cannot be automatically assumed to translate to neurons, we investigated the contribution of NRP-1 to entry in primary PNS sensory neurons. NRP-1 expression was confirmed in SCG, TG, and LS-DRG neurons of hACE2 and WT mice by Western blot (Figure 7a). Immunostaining in LS-DRG neurons showed NRP-1 at cell membranes of neurons and SGCs (Figure 7b). Primary cultured LS-DRG neurons from hACE2 and WT mice were pretreated with EG00229, a selective NRP-1 antagonist, infected with SARS-CoV-2, and viral RNA concentrations were assessed 2 dpi, the first peak for genome replication as determined in our growth curves. Neurons were observed daily for cytopathic effect from treatment and infection; no effects were noted. Viral RNA concentrations were significantly reduced by 99.8% in hACE2 neurons (*p* = 0.0081) and 86.7% in WT neurons (*p* = 0.0141; Figure 7c), indicating that NRP-1 is an entry factor in PNS sensory neurons irrespective of hACE2 expression.

## 3. Discussion

Neurotropic viruses can enter the nervous system hematogenously or through neural pathways. From the blood, they can infect vascular endothelial cells to access underlying tissues, transport across barriers inside extravasating leukocytes, or invade through cytokine-induced disruptions of the blood–brain barrier integrity. Viruses can enter neural pathways through peripheral sensory, autonomic, and/or motor axon terminals and reach the CNS by retrograde transport, often moving trans-synaptically along functionally connected pathways. SARS-CoV-2 likely uses both mechanisms. SARS-CoV-2 CNS invasion has been proposed via infection of OSNs in the nasal neuroepithelium, the olfactory bulb, and its cortical projections [14,15,16,24,26,27,28]. Organoid, stem cell, microfluidic, and mouse models, correlating with human autopsy findings, demonstrate disruption of endothelial barriers and choroid plexus integrity, as well as transcytosis of SARS-CoV-2, supporting hematogenous CNS entry as infection progresses to severe disease that results in death of the human host [5,24,33,34,35,36,37]. While OSNs are a key constituent of the nasal neuroepithelium, the oronasopharynx is innervated by other PNS sensory and autonomic pathways through which SARS-CoV-2 may enter the nervous system. Utilizing hACE2 mice, WT mice, golden Syrian hamsters, and primary PNS neuronal cultures, we show susceptibility of PNS neurons to SARS-CoV-2 infection very early following infection within 18 h of exposure, demonstrating differential replication kinetics and cytopathic outcomes following infection of sensory, autonomic, and central neurons. We also show evidence supporting axonal transport of SARS-CoV-2 from the PNS to the CNS, preceding viremia, through functionally connected neural pathways. Furthermore, we show that NRP-1 is an entry factor for PNS neuronal entry in the absence of hACE2. As COVID-19 neurological symptoms are often related to peripheral neuron dysfunction, focusing solely on CNS neuroinvasion overlooks the potential impacts of SARS-CoV-2 in the PNS.

The detection of SARS-CoV-2 in TGs and SCGs is consistent with their innervation of the oronasal mucosa. Both animal models, mice and hamsters, showed similar viral RNA concentrations at 18 hpi in the TG, SCG, and olfactory bulb, indicating that sensory and autonomic pathways are as susceptible to invasion as the olfactory system. Our results complement findings from post-mortem studies of COVID-19 patients, which have found viral RNA in the TGs of 14% of patients assessed with axonal damage and neuron loss in the TGs of some patients [37,38]. Our use of in vivo and ex vivo infections demonstrates that sensory trigeminal neurons are both susceptible and permissive to productive infection with SARS-CoV-2, culminating in the release of infectious virus, suggesting the TG can serve as an alternative route for CNS invasion. Increasing reports of acute necrotizing hemorrhagic encephalopathy (AHNE) associated with COVID-19 suggest that neuroinvasion may follow a similar pathway as another neurotropic virus, HSV-1, which causes a nearly identical complication upon reaching the CNS from TGs. Invasion of the brainstem along projections of the TG could damage nuclei important in cardiorespiratory regulation, a feature of severe COVID-19, and at least one imaging study of a COVID-19 patient with AHNE suggested invasion of the brainstem via the TG [7]. While viral RNA and proteins have been detected in salivary glands of COVID-19 patients, and infectious virus has been recovered from saliva, no assessments of the SCG have been made in human autopsies [39,40,41,42]. The SCG provides sympathetic nervous system innervation to the oronasal mucosa, salivary and lacrimal glands, and oronasopharyngeal vasculature, providing possible neural entry sites. Retrograde axonal transport along the SCG would allow for invasion of preganglionic neurons in the cervical spinal cord, shown to harbor SARS-CoV-2 RNA and protein at autopsy in some COVID-19 patients [43]. Our neuronal culture infection data indicate abortive infection in the SCG with no production of infectious virus, suggesting that it may be of less concern for direct CNS invasion. However, the pathology of the SCG following infection in vivo in both animal models suggests that SARS-CoV-2 causes significant, and perhaps irreparable, damage to sympathetic neurons. Although we did not examine thoracic ganglia or the sympathetic trunk, the cytopathology in the SCG has broader implications for cardiac function, which relies on autonomic regulation. It is notable that comparable viral RNA concentrations were detected in TGs and SCGs of WT mice and hamsters, indicating that both tissues are equally susceptible to infection in the absence of hACE2.

Entry into TGs and SCGs following intranasal inoculation was expected, but detection of SARS-CoV-2 in LS-DRGs was unanticipated. Viral RNA in LS-DRGs at levels comparable to TGs in both animal models demonstrates that distal sensory neurons, regardless of their location, are equally susceptible to infection. How the virus reached such distal ganglia is uncertain. Given that viral RNA was detected in LS-DRGs 18 hpi, preceding viremia, hematogenous spread is unlikely. It is noteworthy that detection of viral RNA in the LS-DRG and LS spinal cord occurred asynchronously at 18 hpi. However, virus was detected in both the LS-DRG and LS spinal cord by 3 dpi. All mice and hamsters with early spinal cord infection also had brainstem infection. Thus, entry into the spinal cord appears to follow CNS invasion at the brainstem, but infection of LS-DRG neurons is possible from either the central or peripheral axon terminals. Once in the LS-DRG, results from our primary neuronal cultures indicate that viral replication and release of infectious virus occur cyclically every 48 h. As the virus infected a small percentage of sensory neurons in culture, this pattern likely represents two distinct cycles of productive infection. In vivo, however, the majority of LS-DRG neurons were positive for SARS-CoV-2, suggesting that infection of sensory neurons within the host is more efficient than ex vivo infection of cultured neurons. Although COVID-19 symptoms can include tingling, numbness, and burning in fingers and toes, which are indicative of nociceptor damage or dysfunction, and post-mortem studies of COVID-19 patients have found viral RNA in the sciatic nerve [43], LS-DRG neurons have not previously been assessed for their susceptibility to infection with SARS-CoV-2, either in animal models or in human autopsies. However, ACE2 is expressed by a subset of nociceptors in human DRGs, particularly in LS-DRGs [44]. Development of allodynia in hamsters demonstrates that SARS-CoV-2 infection of peripheral sensory neurons results in sensory dysfunction. Spinal cord involvement is becoming increasingly associated with COVID-19, and a recent review of spinal cord disorders in COVID-19 posited that direct invasion of the cord by SARS-CoV-2 could cause these pathologies [45]. Our results indicate that neurons in the LS-DRG and in the spinal cord are susceptible to SARS-CoV-2 infection, which results in allodynia, providing a rationale for a deeper investigation of the LS-DRG as a site of productive viral infection in COVID-19 and the possibility of directional axonal transport.

Our findings reveal a more detailed picture of SARS-CoV-2 invasion into the brain than previous reports and suggest that the PNS can serve as an alternate route of entry. We detected viral RNA and infectious virus in the brainstem, cortex, and hippocampus before viremia. TG neurons, with axonal projections to both the oronasal epithelium and brainstem, or SCG neurons, with synaptic connectivity to the salivary glands and brainstem, both of which were infected preceding viremia, could deliver virus directly to the CNS. Our detection of viral RNA in PNS ganglia and their functionally connected CNS targets preceding viremia provides a physiological mechanism for direct neuronal invasion. Once in the CNS, our results indicate that viral penetration and replication within the brain is region-specific, suggesting the presence of factors (cell types, synaptic connections, vascularization level) that favor invasion and replication in some regions over others. Our results comport with findings from post-mortem studies of COVID-19 patients that show brain region-dependent variation in viral RNA, subgenomic RNA, and viral protein throughout the olfactory bulb, brainstem, and cortex [6,43,46,47].

Notably, SARS-CoV-2 was detected in the PNS and CNS of WT mice and hamsters, indicating that hACE2 expression is not a strict requirement for neuronal infection. While NRP-1 has been shown to facilitate entry of SARS-CoV-2 in various non-neuronal tissues, we have shown, using cultured primary adult sensory neurons in vitro, that NRP-1 facilitates viral entry in the PNS, thus providing a molecular mechanism for direct neuronal invasion. Inhibition of NRP-1 in cultured LS-DRG neurons from hACE2 mice reduced infection to a greater extent than in WT neurons, indicating that NRP-1 can serve as an entry factor to enhance infection in the presence of hACE2 expression or an alternative entry factor independent of hACE2. It is worth noting that viral entry into sensory neurons was not completely ablated by our blockade of NRP-1. The reduction of SARS-CoV-2 RNA copy number by ≈87% in WT sensory neurons pretreated with our NRP-1 inhibitor suggests the potential for additional SARS-CoV-2 entry factors in the PNS, which will be the focus of future studies.

The impact of SARS-CoV-2 on the nervous system is only beginning to be understood, with sensory and autonomic disorders lasting well beyond initial infection, indicating the need for a comprehensive understanding of its impact on the entire nervous system, not just the brain. Case reports/series have described post-COVID-19 syndrome symptoms involving the TG (trigeminal neuralgia) [48,49], SCG (Horner syndrome) [50], LS-DRG (radicular pain) [51], and spinal cord (transverse myelitis) [52]. Our in vitro and in vivo data indicate that these tissues are susceptible to infection with SARS-CoV-2 very early following infection, with variable cytopathic outcomes, and merit investigation. No study has yet assessed the permissiveness of TG or SCG neurons to productive infection by SARS-CoV-2 or investigated their role in viral invasion of the CNS, though some post-mortem studies have detected viral RNA after protracted disease [37,38]. Our revelation of the susceptibility of the TG and SCG to infection may provide insight into clinical observations, such as that TG neuralgia occurs in up to 8.4% of COVID-19 patients and that pharmacological blockade of the cervical sympathetic chain, containing the SCG, can resolve ageusia and dysgeusia immediately following treatment [53,54]. Additionally, no studies have determined if distal sensory ganglia such as the LS-DRG can support infection, let alone provide an avenue of infection to the spine. We found that PNS sensory and autonomic neurons, supporting glial cells, and spinal cord neurons are susceptible and, in most cases, permissive to productive infection with SARS-CoV-2 via direct neural invasion, independent of hACE2, using NRP-1 as an entry factor. We show that neuroinvasion is not a rare event following SARS-CoV-2 infection and that it occurs well before the onset of symptomatic disease. The rapidity of neuroinvasion, as early as 18 hpi, severely limits the availability of human autopsy samples to investigate early neuroinvasion events, as most samples are collected from patients well into their disease process. The presence of infectious virus in these tissues preceding viremia shows that neuroinvasion occurs early in infection via peripheral neural pathways. Further research into these sites of neuroinvasion is necessary with contemporary SARS-CoV-2 isolates, as COVID-19 transitions from a pandemic to an endemic disease associated with long-term neurological sequelae.

## 4. Materials and Methods

### 4.1. Ethics Statement

This study was approved by the Virginia Polytechnic Institute and State University Institutional Animal Care and Use Committee (Protocol # 20-228, approved 1 February 2021, and Protocol #20-184, approved 2 August 2022). This study was carried out according to the US Department of Agriculture’s Animal Welfare Act and the Public Health Service’s Policy on Humane Care and Use of Laboratory Animals.

### 4.2. Cell Lines

Vero E6 cells (ATCC, CRL-1586), historically isolated from the kidney of a female African green monkey, were used for plaque assays. HEK293 cells (ATCC, CRL-1573), historically isolated from the kidney of a female human fetus, were used for production of positive control for Western blot for hACE2 antibody validation. Morphology was used to authenticate cells. New validated cells were purchased from ATCC every 3 years, propagated from the original vial, and frozen in aliquots. A new aliquot was revived from cryopreservation every 2–3 months (not more than 30 passages) to ensure maintenance of cellular integrity. Cell lines were guaranteed free of mycoplasma contamination by ATCC. None of the cell lines used are listed on the Register of Misidentified Cell Lines curated by the International Cell Line Authentication Committee. Vero E6 cells and HEK293 cells were maintained following standard cell culture protocols using Dulbecco’s Modified Eagle Medium (GenClone, El Cajon, CA, USA) supplemented with 8% fetal bovine serum (Sigma, St. Louis, MO, USA) and 1% penicillin/streptomycin (Corning, Corning, NY, USA) and kept at 37 °C with 5% CO_2_.

### 4.3. Viruses

SARS-CoV-2 isolate USA-WA1/2020 (NR-52281; BEI Resources) was passaged twice using Vero E6 cells (ATCC, CRL-1586) by the Bertke Lab to produce viral stock for mouse and primary neuronal culture infections. USA-WA1/2020 was recovered from an oropharyngeal swab taken from a 35-year-old male in Washington state in January 2020 who was diagnosed with COVID-19 after returning from visiting family in Wuhan, China. Viral stocks were titrated in duplicate using a standard plaque assay on Vero E6 cells with agarose overlay [55]. Viral stocks were sequenced by the VT-Molecular Diagnostics Laboratory (Fralin Biomedical Research Institute, Roanoke, VA, USA). Viral sequence data were deposited in GenBank (Accession# OP934235.1). Viral stocks for hamster infections were produced and titrated in Vero E6 cells using an independent aliquot of SARS-CoV-2 isolate USA-WA1/2020 (BEI, NR-52281) by the Duggal lab.

### 4.4. Mouse Infections

Eight to ten-week-old immunocompetent adult male and female B6.Cg-Tg(K18-ACE2)2Prlmn/J mice (Jackson Laboratory, Bar Harbor, ME, USA, Strain#034860; Stock#034860; RRID:IMSR_JAX:034860) and their wild-type C57BL/6J counterparts (Jackson Laboratory, Strain#000664; Stock#000664; RRID:IMSR_JAX:000664) were used for in vivo studies. Health monitoring reports from the vendor were negative for underlying viral, bacterial, or parasitic infections. Sex was considered during study design, and our findings are applicable to both sexes. Sex was assigned by equal assortment among groups. Sex-based analysis of mouse in vivo infections revealed no significant difference in clinical or virological measures, which is similar to previous reports by others [26,28] and is therefore not reported in this manuscript. Mice were randomly assigned to either the inoculum group or control group, ensuring the groups were age- and weight-matched. Whenever possible, experimenters were blinded to genotype and treatment. Mice were genotyped using tail snip samples and RNA isolated from organs for verification of transgene presence and expression using the Jackson Laboratory assay on DNA/cDNA (Appendix A). Sample sizes for in vivo studies were not statistically calculated, as they were similar to sample sizes used in other SARS-CoV-2 studies using K18-hACE2 mice [14,15,24,26]. Mice were housed at a density of up to 5 mice per cage in biocontainment caging held on individually ventilated caging racks (Allentown, Inc., Allentown, NJ, USA) in an on-campus AAALAC-accredited animal BSL3 facility. Mice were provided standard chow and water ad libitum. The light cycle was 12 h of light/12 h of dark. Mice were monitored daily. Mice were acclimated to the environment for one day prior to the study. K18-ACE2 (*n* = 12, 2 groups of 6 mice) and WT mice (*n* = 12, 2 groups of 6 mice) were inoculated intranasally with SARS-CoV-2 isolate USA-WA1/2020 (Appendix A). Inoculations were carried out under ketamine/xylazine anesthesia in the on-campus ABSL-3 facility after a one-day acclimation period. Mice received 20 µL of either 10^3^ PFU (2 groups per mouse type) or 10^5^ PFU (2 groups per mouse type) of SARS-CoV-2 in 1X PBS. The inoculum was split between the nares for each mouse. Uninfected K18-hACE2 control mice (*n* = 2) and C57BL/6J wild-type control mice (*n* = 2) were housed in a separate on-campus ABSL-1 facility. Aliquots of the inocula and viral stock were saved for back titration using plaque assay for infectious viral titer and RT-qPCR for RNA copy number. All mice were genotyped using tail snip following Jackson Laboratory protocol # 38170 V2 (Appendix A). Mice were assessed daily for signs of disease and changes in weight and temperature. Mice from each group (K18-hACE2, WT) and inoculum dose (10^5^ PFU, 10^3^ PFU) were euthanized 3 days post infection (dpi) (*n* = 3) and at 6 dpi (*n* = 3). Tissues collected included blood, CNS tissues (olfactory bulb, hippocampus, brainstem, cerebellum, cortex, spinal cord), PNS tissues (autonomic ganglia: superior cervical ganglia-SCG; sensory ganglia: lumbosacral dorsal root ganglia-LS-DRG, trigeminal ganglia-TG), and viscera (lung). Half of the tissues were collected in TRI Reagent for RNA extraction and RT-qPCR and the other half collected in 10% formalin for immunostaining. Brains were split into hemispheres maintaining attachment with the olfactory bulb. One hemisphere was fixed in formalin for immunostaining and the other dissected out into individual brain regions with each placed in TRI Reagent for RT-qPCR. This experiment was repeated as above for reproducibility, with the addition of an extra mouse in the 10^5^ PFU hACE2 6 dpi group to account for loss due to death. Tissue collection from the second experiment was split between TRI Reagent for RT-qPCR as above or flash frozen on dry ice for plaque assay. Blood samples from the initial and replicate infection study were assessed via RT-qPCR to assess for disseminated infection at 3 and 6 dpi. Lungs from the initial and replicate infection studies were assessed to verify infection at 3 and 6 dpi.

To assess viral spread through nervous tissues at earlier timepoints during infection, and to determine the role that viremia plays verses direct neuronal invasion, K18-hACE2 (*n* = 10) and WT mice (*n* = 10) were infected with 10^5^ PFU SARS-CoV-2 as described above and were euthanized 18 and 42 hpi (*n* = 5 of each group/day). Blood, PNS tissues, CNS tissues (with addition of pituitary gland), and salivary glands were collected as described above for RT-qPCR, plaque assays, and immunostaining.

### 4.5. Golden Syrian Hamster Infections and Von Frey Assessment for Allodynia

Eight-week-old immunocompetent adult male wild-type golden Syrian hamsters (Envigo, Indianapolis, IN, USA, Strain: HsdHan^®^:AURA, Stock #089; RRID:NCBItaxon_10036) were used for in vivo studies. Health monitoring reports from the vendor were negative for underlying viral, bacterial, or parasitic infections. Male hamsters were used when conducting in vivo studies given their documented increased susceptibility to infection and more severe pathology [56,57,58]. Hamsters were randomly assigned to either the inoculum group or control group, ensuring the groups were age- and weight-matched. Sample sizes for in vivo studies were not statistically calculated as they were similar to sample sizes used in other SARS-CoV-2 studies using golden Syrian hamsters [59,60,61]. Hamsters were housed at a density of 2 hamsters per cage in biocontainment caging held on individually ventilated caging racks (Allentown, Inc.) in an on-campus AAALAC-accredited animal BSL3 facility. Hamsters were provided standard chow and water ad libitum. The light cycle was 12 h of light/12 h of dark. Hamsters were monitored daily. To verify that results from the early timepoint neural invasion studies in mice were reproducible in an alternative animal model, and to assess the functional impact of infection on somatosensation, golden Syrian hamsters were infected using an independently propagated viral stock. After a two-day acclimation period in the ABSL-3, hamsters were infected intranasally with 10^5^ PFU of SARS-CoV-2 isolate USA-WA1/2020 (*n* = 9) under isoflurane anesthesia as previously described (Appendix A) [62]. An uninfected control hamster (*n* = 1) was housed in the ABSL-3 facility. Hamsters were monitored daily as described for mice. To assess for allodynia following infection of sensory neurons (LS-DRGs), which mediate transmission of sensory information from the periphery to the CNS, hamsters were tested daily using von Frey filaments on the hind paw footpad. Sensory testing began one day prior to infection, occurred on the day of infection preceding inoculation, and then occurred daily until euthanasia. A series of increasing caliber von Frey filaments were used on the footpad until a withdrawal reflex was elicited. von Frey filaments were cleaned between each animal. Post infection measurements were averaged for each animal and compared to the average of the pre infection measurements of the same animal by analysis of area under the curve. Hamsters were euthanized 18 hpi, 42 hpi, and 3 dpi (*n* = 3 each day) as described above for mice. Tissue types collected, methods of collection, and downstream assays were the same as described for the early timepoint neural invasion studies in mice, with the addition of lacrimal glands.

### 4.6. Primary Neuronal Culture of Peripheral Sensory and Sympathetic Neurons

Eight to ten-week-old immunocompetent adult male and female K18-hACE2 (Jackson Laboratory, Strain#034860, Stock#034860) and WT mice (Jackson Laboratory, Strain#000664, Stock#000664) were euthanized with CO_2_ and transcardially perfused with cold, calcium- and magnesium-free phosphate-buffered saline (PBS). Sensory trigeminal ganglia (TG), sensory lumbosacral dorsal root ganglia (DRG), and sympathetic superior cervical ganglia (SCG) were removed and incubated at 37 °C for 20 min in papain (≥100 units) (Worthington) reconstituted with 5 mL of Neurobasal A medium (Gibco, Waltham, MA, USA) followed by 20 min in 4 mg/mL collagenase and 4.67 mg/mL dispase (Worthington, Columbus, OH, USA) diluted in Hank’s balanced salt solution (HBSS) (Gibco) on a rotator. Following mechanical trituration with a 1 mL pipette, the TG cell suspension was layered on a 4-step Optiprep (Sigma) gradient. Optiprep was first diluted with 0.8% sodium chloride (50.25:49.75 mL) to make a working solution (specific gravity 1.15). Then, the cold Optiprep working solution was diluted with 37 °C Neurobasal A medium to make each step of the gradient: first or bottom layer: 550 µL Neurobasal A medium and 450 µL Optiprep working solution; second layer: 650 µL Neurobasal A medium and 350 µL Optiprep working solution; third layer: 750 µL Neurobasal A medium and 250 µL Optiprep working solution; and the fourth or top layer: 850 µL Neurobasal A medium and 150 µL Optiprep working solution. The gradient with the TG cell suspension was centrifuged at 800× *g* for 20 min, and the two middle layers containing neuronal cells (~2 mL) were collected. SCGs were triturated with a 200 µL pipette in Neurobasal A medium, and DRGs were triturated with a 1000 µL pipette in Neurobasal A medium; neither SCGs nor DRGs went through a gradient step because they contain less ganglionic debris after dissociation. TG, SCG, and DRG were washed twice with Neurobasal A medium supplemented with 2% B-27 (Gibco) and 1% penicillin-streptomycin (Corning). Neurons were counted and plated on Matrigel-coated (Corning) Lab-Tek II chamber slides (ThermoFisher, Waltham, MA, USA) at a density of 3000 neurons/50 µL per well and incubated in a 37 °C/5% CO_2_ incubator in Complete Neuro media (Neurobasal A media supplemented with 2% B-27 (StemCell Technologies, Danvers, MA, USA), 10 mg/mL penicillin-streptomycin (Corning), 10 µL/mL Glutamax (ThermoFisher), 10 µg/mL nerve growth factor (Peprotech, Waltham, MA, USA), 10 µg/mL glial-derived neurotrophic factor (Peprotech), 10 µg/mL neurturin (Peprotech), and 200 mM 5-fluorodeoxyuridine (Sigma)) to deplete non-neuronal cells). After one hour, debris were removed by pipette, and cells were maintained in 500 µL of fresh Complete Neuro media until infection. This primary neuron culture protocol has previously been validated [63,64,65,66,67,68,69,70,71]. Neuronal cultures were allowed to recover for 3 days ex vivo prior to infection. Mice were spot genotyped using the Jackson Laboratory assay for verification of transgene expression using cDNA from a sample from each pooled ganglia taken before plating into replicates from each neuronal culture session (Appendix A).

### 4.7. Primary Neuronal Culture Infection

Neurons from K18-hACE2 and WT mice were inoculated with SARS-CoV-2 isolate USA-WA1/2020 at 30 MOI in 100 µL Neurobasal A (Invitrogen, Waltham, MA, USA) for 8-well chamber slides and 200 µL for 24-well plates for 1 h. Following 1 h of adsorption, the inoculum was removed, fresh complete media was added (without fluorodeoxyuridine), and neurons were incubated at 37 °C with 5% CO_2_. Aliquots of the inocula and viral stock were saved for back titration using plaque assays and RT-qPCR.

### 4.8. Plaque Assays

Flash frozen tissues were homogenized in Dulbecco’s Modified Eagle Medium (DMEM; Fisher Scientific) in bead tubes using a TissueLyser II (Qiagen, Germantown, MD, USA) for 45 s sessions for three sessions. Ganglia were homogenized in 0.5 mL DMEM due to size. Brain regions and spinal cord segments were homogenized in 1.0 mL of DMEM. The undiluted tissue homogenate as well as a ten-fold dilution of homogenate was inoculated in duplicate onto confluent monolayers of Vero E6 cells in 24-well plates. After 1 h of adsorption, the inoculum was removed, a 0.5% agarose overlay was added (DMEM with 8% fetal bovine serum, 1% penicillin/streptomycin, molecular grade agarose), and plates were returned to the incubator for 48 h at 37 °C with 5% CO_2_. Plates were then fixed with 10% formaldehyde, the agarose overlay was removed, and they were stained with plaque dye. Infectious viral titer is reported as plaque forming units per mL (PFU/mL) of tissue homogenate.

### 4.9. RNA Extraction and SARS-CoV-2 Specific RT-qPCR

RNA was extracted and RT-qPCR performed as previously described [72]. Briefly, tissues were homogenized in 200 µL TRI Reagent (Fisher Scientific) using a handheld tissue homogenizer with sterile pestles (Cole-Parmer). RNA was extracted using a standard guanidinium thiocyanate-phenol-chloroform extraction. RNA purity and quantity were assessed using a NanoDrop 2000 spectrophotometer (ThermoFisher). SARS-CoV-2 RT-qPCR reactions (10 µL) using the iTaq Universal Probe One-Step Kit (BioRad, Hercules, CA, USA) and SARS-CoV-2 N1 primers/probe mix (Stock# 10006713; Integrated DNA Technologies, Waltham, MA, USA) were run on a ViiA 7 Real-Time PCR system (Applied Biosystems, Waltham, MA, USA) as described in the instructions for use of the CDC 2019-Novel Coronavirus (2019-nCoV) Real-Time RT-PCR assay. Cycle conditions were as follows: standard setting; 50 °C (10 min, 1 cycle), 95 °C (2 min, 1 cycle), followed by 95 °C (30 s) and 55 °C (3 s) for 45 cycles. Results were reported as genome copy number per 200 ng total RNA. Negative controls for RT-qPCR are reported in Appendix A.

### 4.10. Immunofluorescence

Tissues were prepared for immunostaining as previously described [73]. Briefly, viscera were fixed in 10% formalin and ganglia in 4% paraformaldehyde overnight, moved to 30% sucrose overnight, and subsequently embedded in optimal cutting temperature (OCT) media (ThermoFisher). A Leica CM3050-S cryostat (Leica Biosystems, Nussloch, Germany) was used to prepare 7 µm sections from each tissue block. Slides were rinsed in 1X PBS then blocked in 3% normal donkey serum, 0.1% Triton-100×, and 1× PBS for 30 min at room temperature. SARS-CoV-2 N protein was visualized using an Alexa Fluor^®^ 488 conjugated rabbit monoclonal anti-SARS-CoV-2 nucleocapsid antibody at a 1:1000 concentration (NBP2-90988AF488; Novus Biologicals, Minneapolis, MN, USA). SARS-CoV-2 spike protein was visualized using a mouse monoclonal anti-SARS-CoV-2 spike antibody at a 1:1000 concentration (GTX632604; GeneTex, Irvine, CA, USA) followed by an Alexa Fluor^®^647 conjugated donkey anti-mouse polyclonal antibody at a 1:1000 concentration (ab150111; Abcam, Cambridge, UK). dsRNA was visualized using a mouse monoclonal anti-dsRNA antibody at a 1:500 concentration (MABE1134-25UL; Sigma) followed by an Alexa Fluor^®^647 conjugated donkey anti-mouse polyclonal antibody at a 1:1000 concentration (ab150111; Abcam). hACE2 was visualized using a mouse monoclonal anti-ACE2 antibody at a 1:500 concentration (sc-390851; Santa Cruz Biotechnology, Dallas, TX, USA) followed by an Alexa Fluor^®^647 conjugated donkey anti-mouse polyclonal antibody at a 1:1000 concentration (ab150111; Abcam). NeuN was visualized using an Alexa Fluor^®^ 647 conjugated rabbit monoclonal anti-NeuN antibody at a 1:1000 concentration (ab190565; Abcam). α-d-galactose carbohydrate residues on sensory neurons were visualized using the *Bandeiraea simplicifolia* isolectin B4 (IB4) conjugated to rhodamine at a 1:250 concentration (RL-1102; Vector Laboratories, Newark, CA, USA). Tyrosine hydroxylase was visualized using an Alexa Fluor^®^ 594 conjugated mouse monoclonal anti-TH antibody at a 1:500 concentration (818004; Biolegend, San Diego, CA, USA). Glutamine synthetase was visualized using a mouse monoclonal anti-GS antibody at a 1:100 concentration (MA5-27750; Invitrogen) followed by an Alexa Fluor^®^ 594 conjugated goat anti-mouse polyclonal antibody at a 1:1000 concentration (A11005; Invitrogen). Neuropilin-1 was visualized using a goat polyclonal anti-NRP-1 antibody at a 15 µg/mL concentration (AF566; R&D Systems, Minneapolis, MN, USA) followed by an Alexa Fluor^®^ 647 conjugated donkey anti-goat polyclonal antibody at a 1:1000 concentration (ab150135; Abcam). S100 beta was visualized using an Alexa Fluor^®^647 conjugated rabbit monoclonal anti-S100 beta antibody at a 1:1000 concentration (ab196175; Abcam). Iba1 was visualized using an Alexa Fluor^®^ 647 conjugated rabbit monoclonal anti-Iba1 antibody at a 1:1000 concentration (ab225261; Abcam). Antibodies were validated in house as described below (Appendix A). Nuclei were visualized with 4′,6-diamidino-2-phenylindole (DAPI) in SlowFade Diamond antifade mounting medium (ThermoFisher). Primary antibodies were incubated with tissues overnight at 4 °C in 1% normal donkey serum, 0.1% Triton-100×, and 1× PBS. Secondary antibodies were incubated with tissues for 1 h at room temperature. Mouse-on-mouse interference was reduced using Affinpure donkey polyclonal anti-mouse IgG Fab fragments at a 1:20 concentration (715-007-003; Jackson ImmunoResearch Laboratories Inc., West Grove, PA, USA) for two hours preceding incubation with the unconjugated primary antibodies targeting dsRNA and SARS-CoV-2 spike as described above.

### 4.11. Antibody Validation

The anti-SARS-CoV-2 nucleocapsid antibody (NBP2-90988AF488) was validated by Novus Biologicals via Western blotting. It was validated in our lab through immunostaining for SARS-CoV-2 nucleocapsid in SCGs and TGs (Figure 1 and Appendix A), DRGs and spines (Figure 2, Appendix A), and brains (Figure 3, Appendix A) from infected mice as well as uninfected SCGs, TGs, DRGs, spines, and brains from negative control mice. It was also validated via immunostaining of SCGs, TGs, DRGs, and brains from infected and uninfected negative control hamsters (Figure 6 and Appendix A). It was additionally validated for in vitro use by staining infected and control primary neuronal cultures of SCGs, TGs, and DRGs (Figure 4c–e and Appendix A). Immunostaining was assessed for punctate cytoplasmic staining of viral protein in infected tissue with absence in uninfected tissue. It was further validated in our lab by Western blotting using homogenized SARS-CoV-2 WA1/2020 viral stock prepared for this study (Appendix A). The blot was probed with the unconjugated parental antibody and an HRP conjugated goat polyclonal anti-rabbit IgG secondary antibody (ab6721; Abcam; Lot # GR34222167, Clone: polyclonal), and a band was observed just below 50 kDa, which is in line with the predicted molecular weight of 46 kDa, which is consistent with other published Western blots for this protein [74,75,76,77]. Additional bands were noted between 37 kDa-25 kDa and 25 kDa–20 kDa, which is consistent with other validation Western blots published by vendors for this protein as well as manuscripts listed above (Cell Signaling Technologies, Novus, Rockland, Proteintech, San Diego, CA, USA). The anti-SARS-CoV-2 spike protein antibody (GTX632604) was validated by GeneTex by immunostaining, Western blotting, ELISA, and immunoprecipitation. It was validated in our lab through immunostaining for SARS-CoV-2 spike in SCGs and TGs (Figure 1 and Appendix A) as well as DRGs (Figure 2 and Appendix A) from infected mice as well as uninfected SCGs, TGs, and DRGs from negative control mice. Immunostaining was assessed for punctate cytoplasmic staining of viral proteins in infected tissue with the absence in uninfected tissue. It was further validated in our lab by Western blotting using homogenized SARS-CoV-2 WA1/2020 viral stock prepared for this study (Appendix A). The blot was probed with the antibody and an HRP conjugated donkey polyclonal anti-mouse IgG secondary antibody (PA1-28664; Invitrogen; Lot # WI3375112, Clone: polyclonal), and a band was observed between 250 and 150 kDa, which is in line with the molecular weight of a full-length spike (180–200 kDa), which is consistent with other published Western blots for this protein [78,79,80]. The anti-dsRNA antibody (MABE1134-25UL), a marker of viral genome replication of RNA viruses, was validated by Sigma by immunostaining. It was validated in our lab through immunostaining for dsRNA in SCGs and TGs (Figure 1 and Figure 2) as well as DRGs (Figure 2 and Appendix A) from infected mice as well as uninfected SCGs, TGs, and DRGs from negative control mice. Immunostaining was assessed for punctate cytoplasmic staining of dsRNA in infected tissue with absence in uninfected tissue. As this antibody is specific for RNA, Western blotting was not possible. The anti-hACE2 antibody (sc-390851), a cell membrane protein used by SARS-CoV-2 as a viral receptor, was validated by Santa Cruz via Western blotting. It was validated in our lab through immunostaining of tissues from K18-hACE2 mice vs. wild-type non-hACE2-expressing mice. Immunostaining was assessed for punctate membrane staining of hACE2 in tissue from K18-hACE2 mice with absence in wild-type non-hACE2-expressing mice. It was further validated in our lab by Western blotting using a whole cell homogenate of HEK293 cells as a positive control (Appendix A). It was further used to assess expression of hACE2 in a whole tissue homogenate of SCGs, TGs, and DRGs from K18-hACE2 mice (Figure 6d). The blot was probed with the unconjugated parental antibody and an HRP-conjugated donkey polyclonal anti-mouse IgG secondary antibody (PA1-28664; Invitrogen; Lot # WI3375112, Clone: polyclonal), and a band was observed between 100 and 150 kDa, which is in line with the predicted molecular weight of 100–110 kDa, which is consistent with other published Western blots for this protein [81]. A mild elevation of ACE2 in the positive control (HEK293 cell homogenate) above 100 kDa was observed but was within the range of the molecular weight. It is not uncommon for ACE2 in the HEK293 cell homogenate to appear mildly elevated above 100 kDa on Western blots, likely due to modifications in kidney epithelial cells that are not present in ganglia (R&D Systems). The anti-NeuN antibody (ab190565), a neuronal nuclear protein used as a marker of neurons, was validated by Abcam by Western blotting of the parental antibody. It was validated in our lab through immunostaining of neuronal tissues, including SCGs and TGs (Figure 1 and Appendix A), DRGs and spines (Figure 2, Appendix A), and brains (Figure 3, Appendix A) from infected mice as well as uninfected SCGs, TGs, DRGs, spines, and brains from negative control mice. It was also validated via immunostaining of SCGs, TGs, DRGs, and brains from infected and uninfected negative control hamsters (Figure 6 and Appendix A). Immunostaining was assessed for uniform nuclear staining with minimal cytoplasmic staining. It was further validated in our lab by Western blotting using mouse whole brain homogenate (Appendix A). The blot was probed with the antibody and an HRP-conjugated goat polyclonal anti-rabbit IgG secondary antibody (ab6721; Abcam; Lot # GR34222167, Clone: polyclonal), and doublet bands were observed flanking 50 kDa, which is in line with the molecular weight of 48 kDa, which is consistent with other published Western blots for this protein [82,83,84,85]. The anti-neuropilin-1 antibody (AF566), a surface glycoprotein important in axon growth that binds VEGF in conjunction with tyrosine kinase receptors and is a putative SARS-CoV-2 receptor/co-receptor, was validated by R&D Systems by Western blotting of knockdown samples. It was further validated in our lab by Western blotting using a mouse whole brain homogenate as a positive control (Appendix A). It was further used to demonstrate NRP-1 expression, via Western blot, in a whole tissue homogenate of SCGs, TGs, and DRGs from K18-hACE2 and wild-type mice (Figure 7a and Appendix A). The blots were probed with the antibody and an HRP-conjugated donkey polyclonal anti-goat IgG secondary antibody (PA1-28664; Invitrogen; Lot # WI3375112, Clone: polyclonal), and a single band was observed between 100 kDa and 75 kDa, which is in line with the reported molecular weight of 150–80 kDa, which is consistent with other published Western blots for this protein [86,87,88,89]. Additional bands were noted at ≈250 kDa and between 150 kDa and 100 kDa, which is consistent with other validation Western blots published by vendors for this protein as well as manuscripts listed above, which indicates recognition of both C and N-termini of the protein in a positive control brain homogenate, which is to be expected as this antibody is polyclonal (Cell Signaling Technologies, R&D, Proteintech, Abcam). Only the single banding pattern between 100 kDa and 75 kDa was present in the ganglia and was therefore used in the validation blot for the antibody and to verify presence in the ganglia. The anti-tyrosine hydroxylase antibody (818004), a cytoplasmic marker of autonomic neurons, was validated by Biolegend by Western blotting and by immunostaining of formalin-fixed paraffin-embedded tissues. It was previously validated in our lab through staining of positive control autonomic ganglia (SCG) vs. negative control sensory ganglia (TG, DRG). Immunostaining was assessed for punctate cytoplasmic staining of TH in autonomic neurons with absence in sensory neurons. It was further validated in our lab by Western blotting using a mouse whole brain homogenate (Appendix A). The blot was probed with the antibody and an HRP-conjugated donkey polyclonal anti-mouse IgG secondary antibody (PA1-28664; Invitrogen; Lot # WI3375112, Clone: polyclonal), and doublet bands were observed between 75 and 50 kDa, which is in line with a molecular weight of 60 kDa, which is consistent with other published Western blots for this protein [90,91,92,93]. The anti-glutamine synthetase antibody (MA5-27750), a cytoplasmic marker of satellite glial cells (SGCs), was validated by Invitrogen by knockout with Western blotting. It was validated in our lab by immunostaining of SGCs in DRGs of infected mice (Figure 2g). It was validated for in vitro use by staining primary neuronal cultures of SCGs (Figure 4c and Appendix A). Immunostaining was assessed for cytoplasmic staining of SGCs and absent staining of neurons. It was further validated in our lab by Western blotting using mouse whole brain homogenate (Appendix A). The blot was probed with the antibody and an HRP-conjugated donkey polyclonal anti-mouse IgG secondary antibody (PA1-28664; Invitrogen; Lot # WI3375112, Clone: polyclonal), and a band was observed between 50 and 37 kDa, which is in line with a calculated molecular weight of 42 kDa, which is consistent with other published Western blots for this protein [94,95,96,97]. IB4 (RL-1102), a lectin that binds membrane-bound α-d-galactose residues and is used as a marker of a sub-population of sensory neurons, was validated by Vector Laboratories by immunostaining positive control tissues. It was validated in our lab by immunostaining primary neuronal cultures of sensory TG and DRG neurons as well as autonomic SCG neurons (Figure 4c,d and Appendix A). Immunostaining was assessed for membrane staining of TGs and DRGs with absent staining of SCGs. Our immunostaining is similar to that which has previously been reported for this lectin [98,99,100,101,102]. As this is a lectin that binds a cell surface carbohydrate, Western blotting was not possible. The anti-S100 beta antibody (ab196175), a cytoplasmic calcium-binding protein used as a marker of astrocytes, was validated by Abcam by immunostaining. It was further validated in our lab by Western blotting using a mouse whole brain homogenate (Appendix A). The blot was probed with the antibody and an HRP-conjugated goat polyclonal anti-rabbit IgG secondary antibody (ab6721; Abcam; Lot # GR34222167, Clone: polyclonal), and a band was observed just above 10 kDa, which is in line with a molecular weight of 11 kDa, which is consistent with other published Western blots for this protein [103,104,105]. The anti-Iba1 antibody (ab225261), a cytoplasmic ionized calcium-binding adapter protein used as a marker of microglia, was validated by Abcam by immunostaining.

### 4.12. Confocal Microscopy and Image Analysis

Imaging was performed using a Leica SP8 scanning confocal microscope. Sections of the ganglia, brain, and spinal cord were imaged with identical laser power and gain settings within each tissue type to account for background immunofluorescence. Cells from in vitro studies were imaged with varying laser powers and/or gains due to the wide range of immunofluorescence observed within given experiments. Images were imported into ImageJ, and contrast and brightness were adjusted identically across all images within tissue types. Three-dimensional models were made using ImageJ (v1.53F51) and SyGlass VR (v1.53F51) imaging software.

### 4.13. Quantification of Infection in Autonomic and Sensory Ganglia in Primary Neuronal Culture and Tissues

To quantify the number of autonomic (SCG) and sensory (DRG) neurons infected per ganglia in vitro, 8-well chamber slides containing SCGs and DRGs from hACE2 and WT mice were fixed with paraformaldehyde at 1, 2, and 3 dpi and stained as described above for the detection of SARS-CoV-2 nucleocapsid. The number of infected neurons from each ganglion were counted for each day, averaged, and reported as the percentage of infected neurons per 500 neurons counted. LS-DRG neurons were chosen as the representative sensory neuron as they had the more dynamic replication kinetics with successive rounds of replication. To quantify the number of autonomic (SCG) and sensory (LS-DRG, TG) neurons infected per ganglia in vivo, tissue sections from hACE2 and WT mice were immunostained as described above for SARS-CoV-2. The number of infected neurons from each ganglion were counted across three tissue sections spanning the ganglion being counted, averaged, and reported as described above.

### 4.14. Detection of SARS-CoV-2 Neuronal Replication and Release in Primary Neuronal Culture

To determine if neurons from K18-hACE2 and WT mice are permissive to infection and release of the infectious virus, neurons were incubated for up to 5 dpi (depending on availability of neurons from the specific ganglia) with daily sampling. To determine the amount of virus bound to neurons vs. that left unbound immediately following incubation with the inoculum, the inoculum and neurons were collected separately and constituted the 0 dpi sample. Daily, media and neurons were collected separately in duplicate (TGs, LS-DRGs) or singularly (SCGs) in 500 µL of LS-TRI Reagent (Fisher Scientific) for RNA extraction and viral genome copy number quantitation via RT-qPCR as described above. Samples were stored at 4 °C until processing. For quantification of viral titer in neurons vs. that released into the media, neurons and media were collected separately in duplicate (TGs, LS-DRGs) or singularly (SCGs). To correct for evaporation of media throughout the time course, the final volume of collected media was brought up 500 µL by adding DMEM prior to plaque assay. Neurons were collected in 500 µL DMEM after scraping with a pipette tip. Samples were immediately stored at −80 °C until processing for plaque assay as described above. Following collection of the media but prior to collection of the neurons in TRI reagent or DMEM, the neurons were gently washed with 500 µL DMEM, which was then discarded, to remove any residual media containing RNA or virus. A similar rinse was performed immediately after the 1 h inoculation to remove any residual inoculum.

### 4.15. Inhibition of SARS-CoV-2 Infection by Neuropilin-1 Blockade in Primary Sensory Neuronal Culture

Primary neuronal cultures of LS-DRGs from K18-hACE2 and WT mice were established as described above. Neurons were pretreated with 100 µM of the NRP-1 antagonist EG00229 (6986; Tocris, Bristol, UK) dissolved in DMSO prior to infection as described for Caco-2 cells [30]. EG00229 putatively blocks binding between the carboxyl-terminal sequence of SARS-CoV-2 S1, which has a C-end rule (CendR) motif, and the extracellular b1b2 CendR binding pocket of NRP-1, which has been suggested as an alternative co-receptor for SARS-CoV-2 in non-neuronal cells [30,31,32]. Neurons were infected as described above. To determine if NRP-1 blockade impacted SARS-CoV-2 entry and therefore subsequent replication in neurons, neurons and media were collected together in LS-TRI Reagent (Fisher Scientific). RNA was isolated and virus replication assessed via RT-qPCR as described above. Samples were collected at initial peak replication times as determined through our previous neuronal growth kinetics studies (LS-DRG; 2 dpi) to assess if theses peaks were blunted or completely inhibited. At 2 dpi, we previously detected ≈1 log10 increase in viral RNA copy number above the quantity of viral RNA post-entry in hACE2 neurons (5 log10 vs. 4 log10) and WT neurons (4 log10 vs. 3 log10), indicating viral replication. The most dramatic reduction in viral RNA, if NRP-1 is an alternative entry factor used by SARS-CoV-2, would be observed at 2 dpi, as this is the first indication of viral replication. This acknowledges the fact that entry is likely not total, allowing some viral entry and replication, although at reduced levels compared to untreated neurons. Infected neurons from K18-hACE2 and WT mice not treated with EG00229 but with an equivalent amount of DMSO, the solvent for EG00229, served as controls.

### 4.16. Detection of hACE2 and NRP-1 in PNS Ganglia

Expression of hACE2 in PNS tissues of K18-hACE2 mice as well as NRP-1 expression in K18-hACE2 and WT mice was assessed by standard Western blotting (Appendix A). In summary, ganglia from SCGs, TGs, and DRGs were collected from each uninfected mouse type and homogenized in a RIPA buffer, protein concentration determined via Bradford assay (BioRad), and 15 µg of total protein loaded into the stacking gel and subjected to SDS-PAGE containing trichloroethanol. HEK293 cell homogenate served as a positive control for hACE2. A mouse whole brain homogenate served as a control for NRP-1. Proteins were transferred to a PVDF membrane using a wet transfer. Blots were imaged for total protein using activation of trichloroethanol in a BioRad Chemidoc gel imager (BioRad). Blots were blocked overnight in a cold room in a solution of 5% milk in TBST with Tween. Primary antibody for hACE2 or NRP-1 as listed above were added to blots and incubated for four hours in a cold room. Blots were washed and then probed with appropriate HRP-conjugated secondary antibodies as listed above for 1 h. Blots were washed and imaged on a BioRad Chemidoc gel imager after applying SuperSignal West Femto Maximum Sensitivity Substrate (Thermo). Western blotting was used to determine the presence of NRP-1 in primary neuronal cultures of uninfected hACE2 and WT mice, as these data have yet to be reported elsewhere. Immunostaining for NRP-1 in primary neuronal cultures of uninfected hACE2 and WT mice was used to demonstrate spatial distribution of NRP-1 across the neurons in culture, as these data have not been previously reported elsewhere and provided spatial context for our detection of NRP-1 in the whole neuron homogenate by Western blot.

### 4.17. Statistics and Reproducibility

Sample sizes were not statistically calculated as they were similar to sample sizes used in other SARS-CoV-2 studies using K18-hACE2 mice [14,15,24,26] or golden Syrian hamsters [59,60,61]. Animals were randomly assigned to either the inoculum group or control group, ensuring the groups were age-matched. Measurements were taken from distinct samples. RT-qPCR and plaque assays were performed in duplicate for each sample when assessing both in vivo and ex vivo infections. RT-qPCR results that fell below the lower limit for the standard curve (8 copies) after normalization were reported as 0 for inclusion in the analysis, and no data were excluded. Neuronal infection studies, both immunostaining as well as plaque assay and RT-qPCR studies, were repeated in three separate experiments, with duplicate samples for each ganglion and timepoint in K18-hACE2 and in duplicate in WT mice. Neuropilin-1 inhibition studies were repeated twice, with duplicate samples for each timepoint in K18-hACE2 and WT mice. Mouse infection studies were repeated as described. Golden Syrian hamster infection studies were conducted as described. All statistical analyses were performed in JMP Pro 16 (SAS Institute, Cary, NC, USA) and confirmed in GraphPad Prism version 8 during figure creation. For statistical analysis, significance was set at *p* < 0.05, calculated as two-tailed. RT-qPCR data were log transformed before analysis to correct for normality of distribution. RT-qPCR data were analyzed using a multifactorial ANOVA. If significance was found, pairwise analysis was performed using Tukey’s honestly significant difference (HSD) post hoc test. Inhibitor studies were analyzed using unpaired two-tailed *t*-tests.

## Figures and Tables

**Figure 1 ijms-25-08245-f001:**
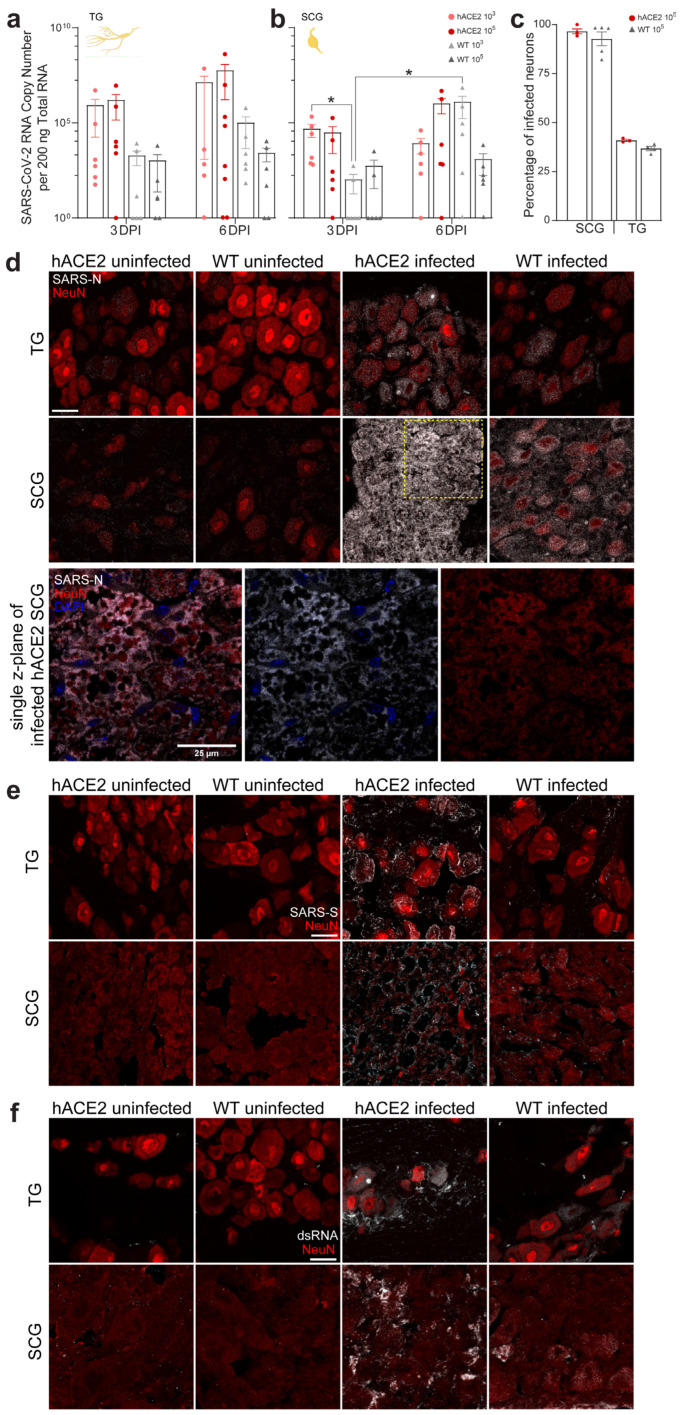
**SARS-CoV-2 infection of TG and SCG in hACE2 and WT mice.** (**a**) SARS-CoV-2 RNA was detected at increasing concentration in TGs of hACE2 and WT mice in both inoculum groups from 3 to 6 dpi. The TG provides sensory innervation to the face, including the nasal septum, and sends projections to the brain stem, thereby providing an alternative entry point for SARS-CoV-2. No statistically significant differences were detected between the groups (F(7, 40) = 1.855, *p* = 0.1032). (**b**) SARS-CoV-2 RNA was detected at increasing concentrations in SCGs of hACE2 and WT mice in both inoculum groups from 3 to 6 dpi. The SCG provides sympathetic innervation to the salivary glands, lacrimal glands, and blood vessels of the head, neck, and brain. Three-way ANOVA detected a significant difference (F(7, 40) = 3.118, *p* = 0.0101) in RNA genome copy number. Tukey’s honestly significant difference (HSD) post hoc tests detected significant differences between the WT groups inoculated with 10^3^ PFU assessed at 3 and 6 dpi (*p* = 0.048). (**c**) Percentage of SCG and TG neurons infected by 6 dpi in tissue sections from 10^5^ PFU-inoculated hACE2 and WT mice. SCGs showed high levels of infection in each mouse type (93–96%) as did TGs (37–41%). (**d**) Immunofluorescence for SARS-CoV-2 nucleocapsid (SARS-N, grey)- and NeuN (red)-labeled neurons in TG and SCG sections at 6 dpi in 10^5^ PFU-inoculated mice. SARS-N was more prevalent in hACE2 than in WT but observable in both. No SARS-N was detected in ganglia from uninfected animals. Neurons in SCG were particularly sensitive to infection; in the magnified single z-plane of the area shown in the yellow box on hACE2-infected SCGs, significant vacuolization can be observed in infected hACE2 SCG cells. Contrast for NeuN was increased in the z-plane to better illustrate residual NeuN immunoreactivity inside SARS-N-negative vacuoles in the SCGs. This cytopathology was common across numerous SCGs in hACE2 mice. (**e**) Immunofluorescence for SARS-CoV-2 spike (SARS-S, grey)- and NeuN (red)-labeled neurons in TG and SCG sections at 6 dpi in 10^5^ PFU-inoculated mice. Immunostaining was similar to that for SARS-N, with greater SARS-S in hACE2 neurons, but present in both. Vacuolization was again observed in SCG neurons. SARS-S was absent in uninfected neurons. (**f**) Immunofluorescence for double stranded RNA (dsRNA, grey)- and NeuN (red)-labeled neurons in TG and SCG sections at 6 dpi in 10^5^ PFU-inoculated mice. Immunostaining for dsRNA, a marker of viral replication, was similar to SARS-S and SARS-N immunostaining in TGs and SCGs. dsRNA was present in greater concentrations in SCGs than TGs and in hACE2 mice than WT mice but was present in both ganglia and both mouse types. Positive dsRNA immunostaining indicates SARS-CoV-2 genome replication in TGs and SCGs. Some nonspecific extracellular binding was observed in uninfected mice, but no intracellular immunofluorescence was observed. Data are the mean ± s.e.m. Log-transformed RNA genome copy numbers were statistically compared by three-way ANOVA (independent variables: inocula, days post infection, genotype). Pairwise comparisons were conducted using Tukey’s HSD post hoc tests. * *p* < 0.05. *n* = 6 for all animals/timepoints/tissues (except 10^5^ hACE2 6 dpi TG: *n* = 7; 10^3^ hACE2 6 dpi TG: *n* = 5). Scale bars are 25 μm. See Appendix A for unmerged images. See Appendix A for additional antibody validation via Western blot, hACE2 genotyping, hACE2 protein expression, and RT-qPCR controls.

**Figure 2 ijms-25-08245-f002:**
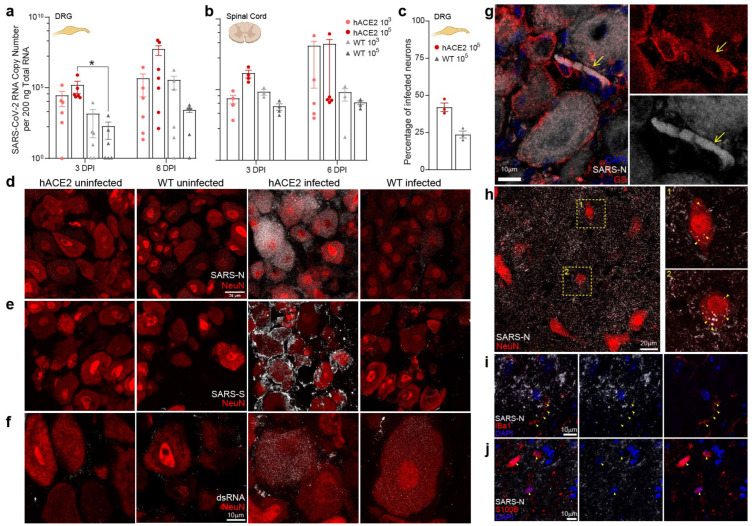
**SARS-CoV-2 infection of LS-DRG, including satellite glial cells, and LS spinal cord in hACE2 and WT mice.** (**a**) SARS-CoV-2 RNA was detected in increasing concentrations in LS-DRGs of hACE2 and WT mice in both inoculum groups from 3 to 6 dpi. The LS-DRG conveys sensory information (pain, pressure, position) from the periphery and organs to the spinal cord. Three-way ANOVA detected a significant difference (F(7, 41) = 4.590, *p* = 0.0007) in RNA genome copy number. Tukey’s HSD detected differences between the hACE2 and WT groups inoculated with 10^5^ PFU at 3 dpi (*p* = 0.018). *n* = 6 for all animals/timepoints (except 10^5^ hACE2 6 dpi: *n* = 7). (**b**) SARS-CoV-2 RNA was detected in lumbosacral spinal cords of hACE2 and WT mice in both inoculum groups at both time points. No statistically significant differences were detected between the groups (F(7, 25) = 1.3054, *p* = 0.2885). Samples sizes were four for 10^5^ hACE2 mice at 3 dpi and five at 6 dpi; five for all timepoints for 10^3^ hACE2 mice; four for all timepoints for 10^5^ WT mice; and three for all timepoints for 10^3^ WT mice. (**c**) Percentage of LS-DRG neurons infected by 6 dpi in tissue sections from 10^5^ PFU-inoculated hACE2 and WT mice. A total of 42% of neurons were infected in hACE2 mice, and 24% were infected in WT mice. (**d**) Immunofluorescence for SARS-N (grey) and NeuN (red) in LS-DRG sections from 10^5^ PFU-inoculated and uninfected hACE2 and WT mice at 6 dpi. SARS-N was more prevalent in hACE2 than in WT but observable in both. No SARS-N was detected in uninfected mice. Detection of RNA and SARS-N in peripheral neurons with no direct connection to the oronasopharynx suggests spread via hematogenous dissemination or via axonal transport. (**e**) Immunofluorescence for SARS-S (grey) and NeuN (red) in LS-DRG sections from 10^5^ PFU-inoculated hACE2 and WT mice at 6 dpi. SARS-S immunostaining was similar to that of SARS-N, with neuronal staining in hACE2 mice and satellite glial cell (SGC) staining in WT mice. (**f**) Immunofluorescence for dsRNA (grey) and NeuN (red) in LS-DRG sections from 10^5^ PFU-inoculated hACE2 and WT mice at 6 dpi. dsRNA immunostaining was similar to that of SARS-N and SARS-S with neuronal staining in hACE2 mice and SGC staining in WT mice. Presence of dsRNA indicated viral genome replication. (**g**) SARS-N (grey) was detected in numerous satellite glial cells (SGCs, glutamine synthetase (GS, red), as denoted by the yellow arrows, surrounding infected LS-DRG neurons at 6 dpi in 10^5^ PFU-inoculated mice. See Appendix A for 3D rendering of this image. (**h**) Representative image of immunofluorescence for SARS-N (grey) and NeuN (red) in spinal cord cross-sections from a 10^5^ PFU-inoculated hACE2 mouse at 6 dpi. SARS-N was observed as discrete puncta in the neuronal cytoplasm, reminiscent of viral replication complexes (arrowheads in h1 and h2, which are magnified areas in the yellow boxes). See Appendix A for 3D rendering of this image (**i**) Representative image of immunofluorescence for SARS-N (grey) and microglial marker Iba1 (red) in spinal cord cross-sections from a 10^5^ PFU-inoculated hACE2 mouse at 6 dpi. Microglia processes are present throughout the cord, as are discrete SARS-N puncta. (**j**) Representative image of immunofluorescence for SARS-N (grey) and astrocyte marker S100B (red) in spinal cord cross-sections from a 10^5^ PFU-inoculated hACE2 mouse at 6 dpi. (Independent variables: inocula, days post infection, genotype). Pairwise comparisons were conducted using Tukey’s HSD post hoc tests. * *p* < 0.05. See Appendix A for unmerged LS-DRGs. See Appendix A for unmerged spinal cord sections and controls. See Appendix A for additional antibody validation via Western blot, hACE2 genotyping, hACE2 protein expression, and RT-qPCR controls.

**Figure 3 ijms-25-08245-f003:**
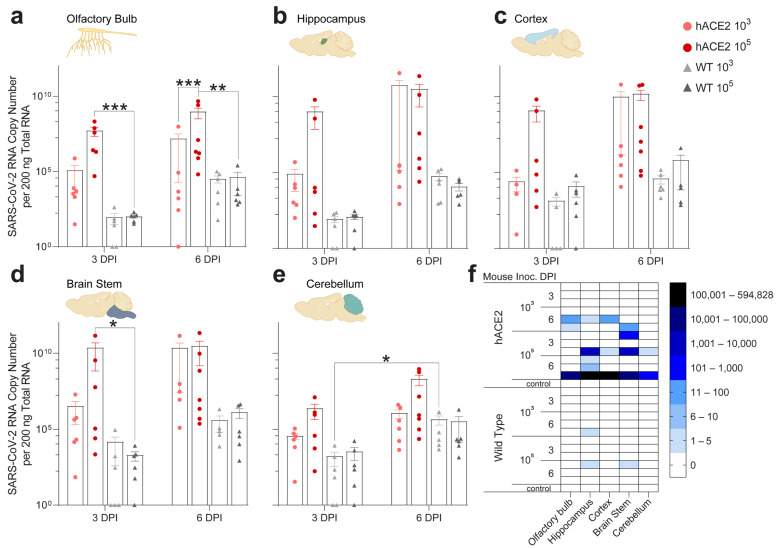
**SARS-CoV-2 infection of the olfactory bulb and various brain regions in hACE2 and WT mice.** SARS-CoV-2 RNA was detected in increasing concentrations from 3 to 6 dpi in the olfactory bulb (**a**), hippocampus (**b**), cortex (**c**), brainstem (**d**), and cerebellum (**e**) of hACE2 and WT mice in both inoculum groups. Three-way ANOVA detected a significant difference (F(7, 41) = 11.825, *p* ≤ 0.0001) in RNA genome copy number in the olfactory bulb. Tukey’s HSD detected differences in the olfactory bulb between hACE2 and WT groups inoculated with 10^5^ PFU assessed at 3 dpi (*p* < 0.0001) as well as between those groups assessed at 6 dpi (*p =* 0.004). A significant difference (F(7, 41) = 5.433, *p* = 0.0002) was also detected in the hippocampi by three-way ANOVA; however, Tukey’s HSD revealed it occurred between non-biologically relevant comparisons. A significant difference (F(7, 41) = 7.217, *p* ≤ 0.0001) was also detected in the brainstem of the hACE2 and WT groups inoculated with 10^5^ PFU assessed at 3 dpi (*p* = 0.0107). While differences were detected in the cortex (F(7, 41) = 6.302, *p* = <0.0001) none were between relevant groups. A significant difference was detected in the cerebellum (F(7, 41) = 6.996, *p* ≤ 0.0001) of the WT groups inoculated with 10^3^ PFU when compared between 3 and 6 dpi. (**f**) Heatmap showing recovery of infectious SARS-CoV-2 from homogenates of the olfactory bulb and specific brain regions assessed for viral RNA. Recovery of infectious virus varied across individual animals, with some having no regions with the recoverable virus and some with the virus in all regions. Of note, infectious virus was recovered from the hippocampi (5 PFU/mg homogenate) and brainstems (3 PFU/mg homogenate) of some WT mice, which are regions functionally impacted in COVID-19 disease. Data are the mean ± s.e.m. Log-transformed RNA genome copy numbers were statistically compared by three-way ANOVA (independent variables: inocula, days post infection, genotype). Pairwise comparisons were conducted using Tukey’s HSD post hoc tests. * *p* < 0.05, ** *p* < 0.01, *** *p* < 0.001. *n* = 6 for all animals/timepoints/tissues (except 10^5^ hACE2 6 dpi brain regions: *n* = 7). See Appendix A for control hACE2 brain sections and Appendix A for infected and control WT brain sections. See Appendix A for additional antibody validation via Western blot, hACE2 genotyping, hACE2 protein expression, and RT-qPCR controls.

**Figure 4 ijms-25-08245-f004:**
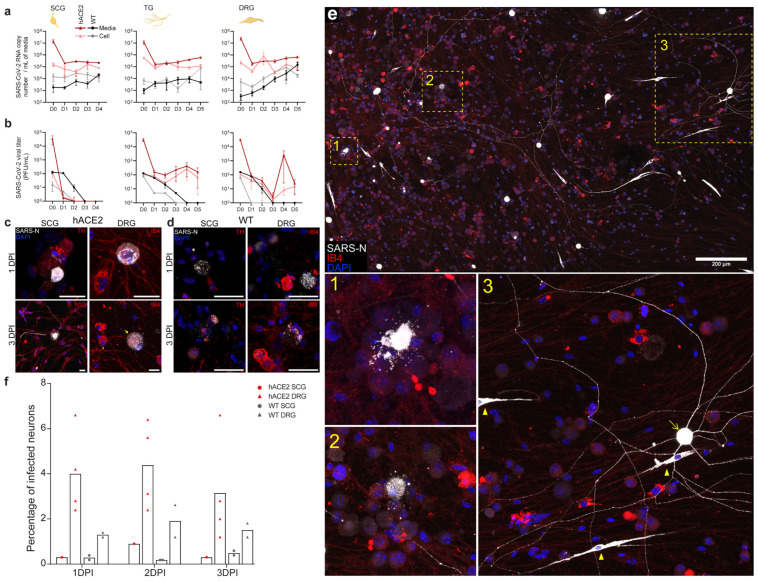
**SARS-CoV-2 infection of primary PNS neuronal cultures from SCG, TG, and LS-DRG of hACE2 and WT mice.** Primary neuronal cultures were generated from SCG, TG, and LS-DRG from 8–10 wk old hACE2 and WT mice. (**a**) SARS-CoV-2 RNA was quantified by RT-qPCR separately in neuronal cells and media to generate a 4–5-day viral genome replication profile. Intracellular replication patterns were similar between hACE2 and WT neurons, although at reduced levels in WT neurons, with increasing viral RNA detected in media over the course of infection. hACE2 LS-DRGs had peaks in genome replication at ~48 hpi and ~96 hpi, indicating successive rounds of replication. Results are from three separate neuronal cultures, each with duplicate technical replicates per ganglion/timepoint. (**b**) Infectious virus was quantified by plaque assay on Vero E6 cells from cellular and media fractions of SCG, TG, and LS-DRG neuronal cultures to generate growth curves in primary neurons from hACE2 and WT mice. Infectious virus was not recovered from SCG neurons, indicating abortive infection, likely mediated by cytotoxicity. Infectious virus was recovered from TG and LS-DRG neurons, indicating productive infection of these neurons, although sustained production of viral progeny is dependent on hACE2. Results are based on sample sizes as reported for above for panel a. (**c**) hACE2 neuronal cultures: immunofluorescence for SARS-N (grey) and either tyrosine hydroxylase (TH, red) or Isolectin-B4 (IB4, red) to counterstain SCG and LS-DRG neurons, respectively, or glutamine synthetase (GS, red) to stain satellite glial cells. SARS-N was observed in neurons from each of the ganglia. Infected neurons were largely free of neurites by 1 dpi. At 3 dpi, many infected neurons exhibited cytopathologies such as degraded neurites, enlarged multi-nucleated cell bodies (arrow) compared to uninfected neurons (arrowhead), and SARS-N+ puncta reminiscent of viral replication compartments. See Appendix A for 3D rendering of LS-DRG at 3 dpi. See Appendix A for 3D rendering of TG at 2 dpi. (**d**) WT neuronal cultures: immunofluorescence for SARS-N (grey) and either TH or IB4 (red) to counterstain neurons or GS (red) to stain satellite glial cells. Immunostaining revealed a similarly heterogenous infection of neurons and satellite glial cells as observed in hACE2 neurons. (**e**) Immunofluorescence for SARS-N (grey) and IB4 (red) to counterstain hACE2 LS-DRG neurons at 3 dpi shows a variety of phenotypes of infected cells, including neurons with a loss of membrane integrity (inset 1), SARS-N+ puncta within and surrounding neurons (inset 2), and seemingly healthy neurons with extensive neurites with strong SARS-N+ staining (arrow in inset 3). Infected satellite glial cells were also observed (arrowheads in inset 3); many appeared to be activated, noted by the presence of extended cellular processes. These findings are similar to immunostaining of LS-DRGs of hACE2 and WT mice in vivo, which also contained numerous infected satellite glial cells. (**f**) Percentage of hACE2 autonomic (SCG) and sensory (LS-DRG) cultured neurons positive for SARS-N were counted from 1 to 3 dpi. A small percentage of autonomic (SCG) neurons were visibly infected, with significant observable cell death, similar to in vivo observations. Infection in sensory (LS-DRG) neurons was consistent from 1 to 3 dpi, with ~5% infected. Infection of neurons ex vivo is less efficient than infection in vivo. Scale bar = 20 μm. Data are the mean ± s.e.m. See Appendix A for unmerged and control images. See Appendix A for additional antibody validation via Western blot, hACE2 genotyping, hACE2 protein expression, and RT-qPCR controls.

**Figure 5 ijms-25-08245-f005:**
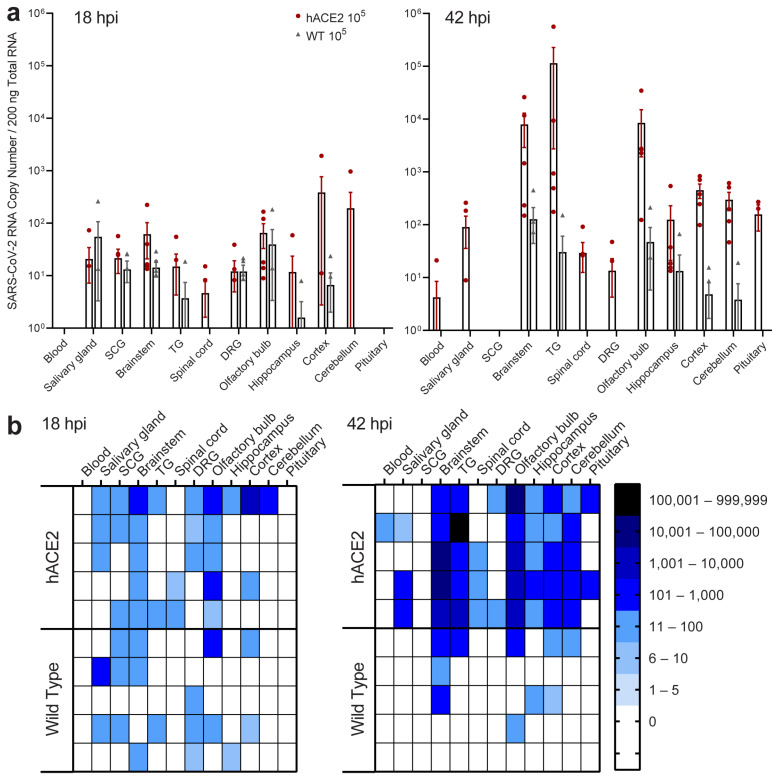
**Pre-viremic neuroinvasion of SARS-CoV-2 into PNS and functionally connected CNS tissues of hACE2 and WT mice as early as 18 h post-infection.** (**a**) Although no viral RNA was detected in blood, low levels of SARS-CoV-2 RNA were detected in PNS and CNS of both hACE2 and WT mice as early as 18 hpi. CNS invasion in hACE2 and WT mice was common in the olfactory bulb, hippocampus, cortex, and brainstem. Viral RNA was detected in PNS ganglia and the tissues they innervate (SCG-salivary gland, TG-brainstem) in both hACE2 and WT mice. Viral RNA was detected separately in the LS-DRGs and spinal cords of some mice. By 42 hpi, viral RNA was detected in blood in only one hACE2 mouse but had increased in the brainstem, TG, and olfactory bulb, indicating replication in these tissues. Sample sizes were hACE2 mice *n* = 10 (5 per timepoint) and WT mice *n* = 10 (5 per timepoint). (**b**) Heatmaps visually displaying RT-qPCR values from panel a. Neuroinvasion in both PNS and CNS occurs rapidly before detectable viremia, thereby indicating direct neural entry and trans-synaptic spread of SARS-CoV-2. Detection of viral RNA in the LS-DRGs but not the spinal cord in some mice and vice versa in others indicates that separate entry routes exist for the LS-DRG and spinal cord. Invasion of the cord likely occurs from the brainstem, as all mice with early spinal cord infection also had brainstem infection. Invasion of the LS-DRG may occur from the periphery. Infectious virus was not detected by plaque assay, indicating the virus had not yet started replicating at these early time points. See Appendix A for hACE2 genotyping, hACE2 protein expression, and RT-qPCR controls.

**Figure 6 ijms-25-08245-f006:**
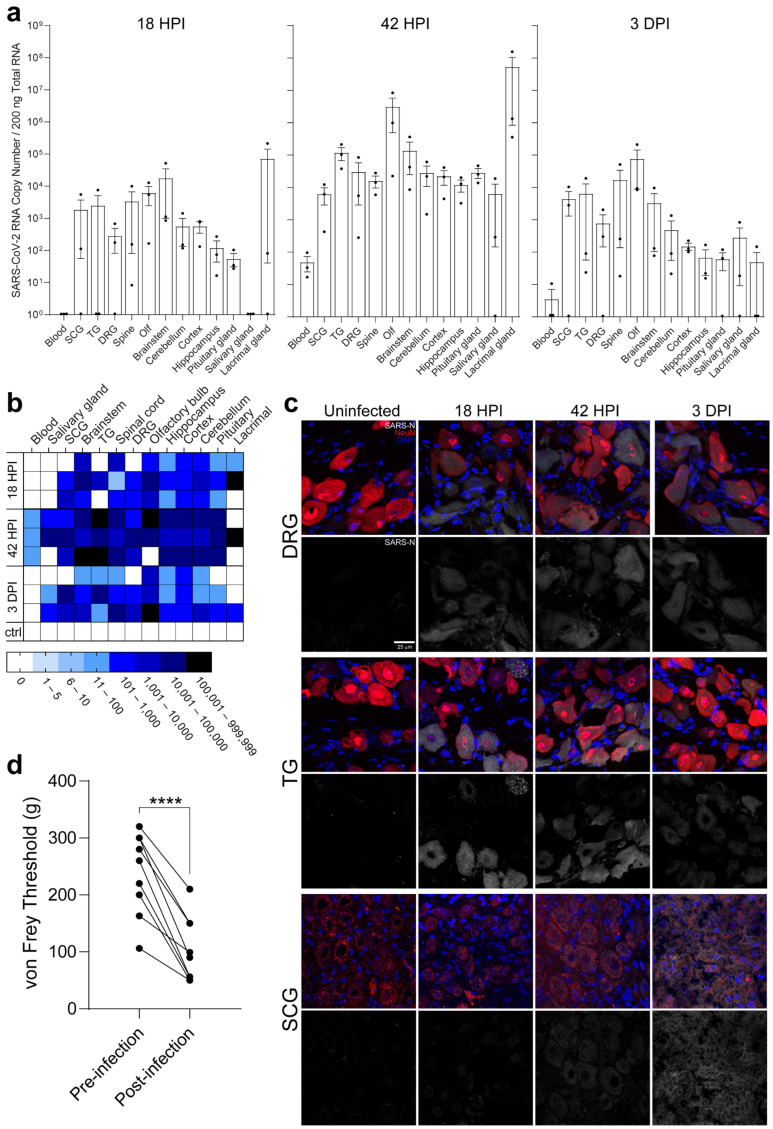
**Pre-viremic neuroinvasion of SARS-CoV-2 into PNS and functionally connected CNS tissues in Syrian golden hamsters as early as 18 h post-infection.** (**a**) Similar to hACE2 and WT mice, no viral RNA was detected in hamster blood at 18 hpi; however, low levels of SARS-CoV-2 RNA were detected in PNS and CNS tissues. Viral RNA was detected in PNS ganglia and the tissues they innervate (SCG-lacrimal gland, TG-brainstem, LS-DRG-spinal cord) as early as 18 hpi. By 42 hpi, viral RNA was increased in all PNS tissues (excluding SCGs), indicating replication in these tissues. Lacrimal glands harbored notably high viral RNA concentrations. Viral RNA copy number was stable in SCGs at 3 dpi (4 log) and had returned to levels observed at 18 hpi in TG and LS-DRGs (4 log and 3 log, respectively), indicating a burst of viral replication. Sample size was *n* = 9 (3 hamsters per timepoint). (**b**) Heatmaps visually displaying RT-qPCR values from panel a. Neuroinvasion in both PNS and CNS occurs rapidly before detectable viremia, thereby indicating direct neural entry and trans-synaptic spread of SARS-CoV-2, as observed in mice. (**c**) Immunofluorescence for SARS-N (grey) and NeuN (red) in LS-DRG, TG, and SCG sections from infected and uninfected hamsters at 18 hpi, 42 hpi, and 3 dpi. SARS-N appears in all ganglia by 18 hpi and continues to be present to 3 dpi. Significant pathology/vacuolization is observed in SCGs at 3 dpi, similar to the pathology observed in mice. (**d**) Results of von Frey threshold test showing a decrease in the amount of force required to elicit a withdrawal reflex, indicating allodynia as a consequence of infection. A paired samples two-tailed t-test found this decrease to be significant (t(8) = 7.606, *p =* 0.0001). Presence of SARS-CoV-2 neuroinvasion of LS-DRGs likely mediates this allodynia, which was noted in some animals (five of nine) as early as 18 hpi and all remaining animals (three of three) by 3 dpi. Scale bar = 20 μm. Data are the mean ± s.e.m. See Appendix A for immunostaining of infected hamster brain. See Appendix A for additional antibody validation via Western blot and RT-qPCR controls. **** *p* < 0.0001.

**Figure 7 ijms-25-08245-f007:**
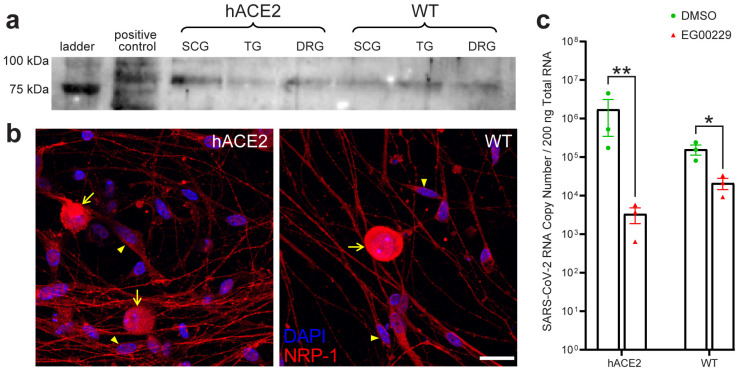
**The role of NRP-1 during entry of SARS-CoV-2 in PNS neurons.** (**a**) Western blot of ganglia (SCG, TG, LS-DRG) from uninfected hACE2 and WT mice showing the presence of the transmembrane glycoprotein neuropilin-1 (NRP-1) on primary neurons and satellite glial cells in culture. NRP-1 has been shown to increase SARS-CoV-2 entry into non-neuronal cells. (**b**) Immunofluorescence for NRP-1 (red) and DAPI (blue) in primary neuronal cultures of LS-DRG neurons from uninfected hACE2 and WT mice, showing that NRP-1 is present across the soma and processes of neurons (arrows) as well as satellite glial cells (arrowheads). Immunostaining for NRP-1 was used to demonstrate spatial distribution of NRP-1 across the neurons in culture as a complement to our detection of NRP-1 in whole neuron homogenate by Western blot. (**c**) Treatment of hACE2 and WT LS-DRG neurons with the NRP-1 antagonist EG00229 prior to infection with SARS-CoV-2 significantly reduced viral RNA concentrations at 2 dpi, the initial peak of viral replication in hACE2 LS-DRG neurons as determined through our replication/growth curves, by 99.8% (t(4) = 4.896, *p* = 0.0081) in hACE2 neurons (DMSO: 1,739,333.3 SARS-CoV-2 RNA copies/200 ng RNA; EG00229: 3,362 SARS-CoV-2 RNA copies/200 ng RNA) and 86.7% (t(4) = 4.165, *p* = 0.0141) in WT neurons (DMSO: 158,933.3 SARS-CoV-2 RNA copies/200 ng RNA; EG00229: 21,233.3 SARS-CoV-2 RNA copies/200 ng RNA). Thus, NRP-1 is an entry factor mediating viral entry into neurons expressing hACE2 and also enhances viral entry into WT neurons. These data also indicate that additional host proteins are involved in neuronal entry. Scale bar = 20 μm. Data are the mean ± s.e.m. Results are from two separate neuronal cultures, each with duplicate technical replicates per ganglion. See Appendix A for additional antibody validation via Western blot, hACE2 genotyping, hACE2 protein expression, NRP-1 protein expression, and RT-qPCR controls. * *p* < 0.05, ** *p* < 0.01.

## Data Availability

Data generated during this study and referenced in this manuscript are available from the corresponding author upon reasonable request.

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
