# Peer review of "SARS-CoV-2 Rapidly Infects Peripheral Sensory and Autonomic Neurons, Contributing to Central Nervous System Neuroinvasion before Viremia"

_ijms, 2024, doi:10.3390/ijms25158245_

Round 1

Reviewer 1 Report

Comments and Suggestions for Authors

The current manuscript by Joyce et al. investigated infection of SARS-CoV-2 in the peripheral nervous system including the sensory ganglia and autonomic ganglia. This is a well written manuscript with several novel points reported. These include the rapid infection of these peripheral ganglia with SARS-CoV-2. The data presented clearly demonstrate the susceptibility of these ganglia to infection with SARS-CoV-2. Another novel aspect of this is the authors reporting infection of these neurons in wild-type mice, that are not susceptible to productive infection with this SARS-CoV-2 variant.

Within the manuscript, additional data and detail is needed:

It should be noted somewhere in the manuscript that the 105 dose of SARS-CoV-2 is likely lethal in all of the hACE2 mice. 1 mouse died by 6 days in the current experiments, but, if extended, most if not all of these mice would have died. This context is important as this represents very different aspects of infection from the low dose in K18-hACE2 mice as well as infection in wild-type mice.

In the Figures, the images should be presented with the control uninfected 1st followed by the images from infected mice.

In the Figure 1 and 2 legend (as well as Figure S2), the dose of virus used for these mice from which the images are obtained needs to be specified.

Assuming that the immunofluorescence images are from the 105 dose of virus, immunofluorescence of the brain and peripheral ganglia for the 103 dose is needed as this typically produces a mild infection in K18-hACE2 mice, with low mortality.

Q-PCR for SARS-CoV-2 from uninfected tissue needs to be included to validate the detection of virus in the infected tissue. Levels are very low, and therefore showing a significantly higher amount is needed to support the data that the neurons in the wild-type mice are actually infected.

In the abstract line 30, it should be specified “we demonstrate, in vitro, that neuropilin-1 facilitates SARS-CoV-2 . . .”

Other edits:

Line 61- PASC not defined

Line 96 -Add “In vitro, we show . . . “ to the role of neuropilin

Line 732 – 37oC

Author Response

We would like to thank the reviewer for their thoughtful comments and suggestions. We have revised our manuscript accordingly and clarified several concerns.

Comment 1: It should be noted somewhere in the manuscript that the 105 dose of SARS-CoV-2 is likely lethal in all of the hACE2 mice. 1 mouse died by 6 days in the current experiments, but, if extended, most if not all of these mice would have died. This context is important as this represents very different aspects of infection from the low dose in K18-hACE2 mice as well as infection in wild-type mice.

Response 1: We agree with the reviewer that context of dose and response is important and as such have added a sentence explaining that the timepoints we assessed were before timepoints where lethality of infection in the hACE2 mice are typically seen. The sentence ”While only one hAEC2 mouse in the 105 PFU inoculated group died during our study, infection likely would have been universally fatal at later timepoints, as significant mortality is commonly reported in hACE2 mice beginning 5-6 dpi” was added at lines in section 2.1 of results from lines 106-109. Our goal was to assess early events following infection, prior to the development of severe disease symptoms, to determine where the virus spread in the nervous system and how it arrived there. These early spread events would be severely impacted upon development of neuroinflammation, breakdown of the blood brain barrier, and influx of immune cells, all of which occur later in infection, leading to the death of the animal.

Comment 2: In the Figures, the images should be presented with the control uninfected 1st followed by the images from infected mice.

Response 2: We have rearranged Figures 1, 2, S2 and S3 to start with uninfected tissues followed by infected tissues. Figure 6 is already in this orientation.

Comment 3: In the Figure 1 and 2 legend (as well as Figure S2), the dose of virus used for these mice from which the images are obtained needs to be specified.

Response 3: We have updated our figure captions for Figures 1, 2, and S2 to specify the dose of virus, 105 PFU/mL, used for these mice.

Comment 4: Assuming that the immunofluorescence images are from the 105 dose of virus, immunofluorescence of the brain and peripheral ganglia for the 103 dose is needed as this typically produces a mild infection in K18-hACE2 mice, with low mortality.

Response 4: The reviewer is correct that the immunofluorescence images for our hACE2 and WT mouse infections are from mice inoculated with 105 PFU. As our goal is to investigate mechanisms of neuroinvasion during early acute SARS-Cov-2 infection, we felt that inclusion of immunostaining images from 105 PFU would best illustrate the location of the virus at these early timepoints in a more robust way than would immunostaining images from mice inoculated with 103 PFU. Also, the localization of virus was similar in 103 PFU-inoculated mice, so the addition of very similar images seemed redundant. Additionally, we included immunostaining from mice inoculated with 105 PFU to allow for comparison with our immunostaining from our golden Syrian hamster infections, which were also at 105 PFU. The inclusion of immunostaining from 105 PFU-inoculated mice also allows for comparison to the seminal work of Oladunni et al. (PMID: 33257679) who used k18-hACE2 mice inoculated with 105 PFU to investigate the distribution of virus and viral induced pathologies throughout this mouse model during the early days of the SARS-CoV-2 pandemic. Several other research groups have published immunostaining images of hACE2 mice infected with 103 PFU and our inclusion would be duplicative, at least with respect for the brain (PMID: 34668775, 35632761, 32841215).

Comment 5: Q-PCR for SARS-CoV-2 from uninfected tissue needs to be included to validate the detection of virus in the infected tissue. Levels are very low, and therefore showing a significantly higher amount is needed to support the data that the neurons in the wild-type mice are actually infected.

Response 5: Supplementary Figure 6F reports the results for negative control tissues from uninfected hACE2 mice (n=2), WT mice (n=2), and hamsters (n=1) for RT-qPCR assay reported in Figures S1C, 1AB, 2AB, 3A-E. Supplementary Figure 6G reports the results for negative control tissues from uninfected hACE2 mice (n=2) and WT mice (n=2) TGs, SCGs, and DRGs used for SARS-CoV-2 growth curves reported in Figure 5A.

Comment 6: In the abstract line 30, it should be specified “we demonstrate, in vitro, that neuropilin-1 facilitates SARS-CoV-2 . . .”

Response 6: We agree that specifying that our neuropilin-1 inhibition studies were conducted in vitro vs in vivo provides proper context. We have amended the sentence as the reviewer requests.

Comment 7: Line 61- PASC not defined

Response 7: We have replaced PASC (post-acute sequelae of SARS-CoV-2 infection) with post-COVID-19 syndrome.

Comment 8: Line 96 -Add “In vitro, we show . . . “ to the role of neuropilin

Response 8: We agree that specifying that our neuropilin-1 inhibition studies were conducted in vitro vs in vivo provides proper context. We have amended the sentence as the reviewer requests.

Comment 9: Line 732 – 37oC

Response 9: We have corrected the superscript error.

Reviewer 2 Report

Comments and Suggestions for Authors

In the present manuscript, the authors found that infection of PNS with SARS-CoV-2 contributes to CNS infection. They also found that NRP-1 was expressed in SCG, TG, and LS-DRG neurons of hACE2 and WT mice, inhibiting NRP-1 by EG00229 significantly decreased the viral load of SARS-CoV-2. This study is very interesting, and the manuscript is well-written.

1.     Figure 7, did the authors see colocalization of NRP-1 and SARS-CoV-2 S or N protein?

2.     Did the authors do NRP-1 knockdown or knockout to test its effect on viral entry to avoid any off-target effect of EG00229?

3.     Figure 7C, they test the effect of EG00229 on viral entry 2 days post-infection which may not be a good assay and time point. Does NRP-1 have functions to help virus replication? They should test the copies of the virus early after the SARS-CoV-2 infection.

Author Response

We would like to thank the reviewer for their thoughtful comments and suggestions. We have revised our manuscript accordingly and clarified several concerns.

Comment 1: Figure 7, did the authors see colocalization of NRP-1 and SARS-CoV-2 S or N protein?

Response 1: The immunostaining images shown in Figure 7B are from uninfected cultures of lumbosacral (LS)-DRG neurons from hACE2 and WT mice. We used immunostaining to identify the distribution of NRP-1 across the neurons in culture as this data had not been previously reported elsewhere. We included NRP-1 immunostaining to demonstrate spatial distribution to complement our detection of NRP-1 in whole neuron homogenate shown in Figure 7A. We have clarified this in the figure caption and methods. The following text has been added to section 4.16 in our methods section “Western blotting was used to determine the presence of NRP-1 in primary neuronal cultures of uninfected hACE2 and WT mice as this data has yet to be reported elsewhere. Immunostaining for NRP-1 in primary neuronal cultures of uninfected hACE2 and WT mice was used to demonstrate spatial distribution of NRP-1 across the neurons in culture as this data has not been previously reported elsewhere and provides spatial context to our detection of NRP-1 in whole neuron homogenate by western blot.” The following text has been added to the figure caption of Figure 7a “uninfected hACE2 and WT mice”. The following text has been added to Figure 7b “Immunostaining for NRP-1 was used to demonstrate spatial distribution of NRP-1 across the neurons in culture as a complement to our detection of NRP-1 in whole neuron homogenate by western blot.”

Comment 2: Did the authors do NRP-1 knockdown or knockout to test its effect on viral entry to avoid any off-target effect of EG00229?

Response 2:Primary adult neurons do not readily uptake transfected siRNA and they respond to lentiviral and adenoviral vectors with an antiviral response, altering the neuronal environment in ways that impact the cellular response to other viral infections, such as SARS-CoV-2. Therefore, we have attempted to knock down NRP-1 using an NRP-1 mRNA targeting vivo-morpholinos, which have been reported to be used successfully with primary neurons. However, the morpholinos produce significant cytotoxicity in our primary adult neuronal cultures, even at very low concentrations. Work is still ongoing and we are seeking an appropriate NRP-1 KO mouse for cultures to further assess the role of NRP-1. The use of a more specific NRP-1 targeting method will be included in a planned follow up manuscript assessing replication kinetics and receptor usage of contemporary SARS-CoV-2 variants vs ancestral SARS-CoV-2. Our data reported in this manuscript will serve to set the stage for our planned follow up studies.

Comment 3: Figure 7C, they test the effect of EG00229 on viral entry 2 days post-infection which may not be a good assay and time point. Does NRP-1 have functions to help virus replication? They should test the copies of the virus early after the SARS-CoV-2 infection.

Response 3: We selected our two-day timepoint based on the data we obtained while generating our SARS-CoV-2 growth curves in hACE2 and WT LS-DRG neurons (Figure 4A). At 2 days post infection, we detected ≈1 log10 increase in viral RNA copy number above the quantity of viral RNA post-entry in hACE2 neurons (5 log10 vs 4 log10) and WT neurons (4 log10 vs 3 log10). An increase of 1 log10 (10-fold increase in viral genomic RNA) in our neuronal cultures is a strong indicator of viral replication. If NRP-1 is indeed an alternative entry factor used by SARS-CoV-2, and if EG00229 could prevent SARS-CoV-2 spike and NRP-1 binding, then the most dramatic reduction in viral RNA would be observed  at 2 dpi, as this is the first indication of viral replication. If we are correct, then viral entry should have been prevented and replication could not occur. However, the blockage of entry from EG00229 was only partial and virus that entered was able to replicate, which we identified at 2 dpi, although the quantity we detected was significantly less than without EG0029. As stated above, we are continuing our investigation on the role of NRP-1 and anticipate a follow up manuscript focused on this topic. The following text has been added to section 4.15 of our methods to clarify the experimental designs “At 2 dpi, we previously detected ≈1 log10 increase in viral RNA copy number above the quantity of viral RNA post-entry in hACE2 neurons (5 log10 vs 4 log10) and WT neurons (4 log10 vs 3 log10), indicating viral replication. The most dramatic reduction in viral RNA, if NRP-1 is an alternative entry factor used by SARS-CoV-2, would be observed at 2 dpi, as this is the first indication of viral replication. This acknowledges the fact that entry is likely not total, allowing some viral entry and replication, although at reduced levels compared to untreated neurons.”

Round 2

Reviewer 1 Report

Comments and Suggestions for Authors

The authors adequately addressed my previous comments.